# iCAS: A In-Context Anomaly Segmentation Framework for Industrial Visual Inspection

## Abstract

Visual in-context prompting has recently made promising progress, achieving training-free segmentation with a generalized model derived from large-scale pre-training. However, we observe that these in-context segmentation models fail on the anomaly detection task, e.g., visual inspection. In this study, we propose iCAS, a novel model for In-Context Anomaly Segmentation enabling automatic defect annotation and visual prompting anomaly segmentation. The framework is built upon an in-context mask transformer, further enhanced by a greedy query selection strategy and a mask-level feature matching module to improve both sensitivity and generalization. Further, we propose the General-to-Specific pre-training to solve the weak generalization problem caused by the scarcity of anomalous samples. Finally, we conduct comprehensive experiments under a variety of anomaly detection and segmentation tasks. Evaluations on multiple publicly available datasets show the generalization and effectiveness of our method.

## 1 Introduction

Recent advances in large-scale pre-trained visual in-context models have led to significant progress in promptable semantic segmentation, demonstrating impressive adaptability across diverse domains and tasks Wang et al. (2023b;a); Kirillov et al. (2023); Zhang et al. (2024a); Meng et al. (2024). However, developing a general anomaly segmentation model for industrial visual inspection remains uniquely challenging. Unlike conventional segmentation tasks, anomaly segmentation in industrial settings demands the precise localization of subtle, fine-grained, and often low-contrast deviations from normal regions Bergmann et al. (2019); Zou et al. (2022). The inherently subtle nature of anomalies, combined with the scarcity of labeled anomaly data, makes it particularly difficult to train models that generalize effectively to diverse and unseen anomalies. Moreover, despite the strong generalization capabilities of recent large-scale pre-trained segmentation models such Kirillov et al. (2023), their representations remain optimized for broad semantic boundaries rather than the subtle, fine-grained deviations characteristic of industrial anomalies. In contrast, an in-context segmentation paradigm enables the model to adapt its behavior dynamically based on a few provided normal or anomalous examples, without the need for full retraining. This paradigm is particularly valuable in industrial inspection, where sample diversity is high and annotation budgets are limited. By leveraging contextual prompts from a small support set, the model can refine anomaly localization, improve boundary precision, and flexibly accommodate novel anomaly types as they arise.

Prior studies on general anomaly segmentation have primarily focused on vision-language models such as CLIP Radford et al. (2021). WinCLIP Jeong et al. (2023) introduces handcrafted text prompts to represent normal and anomalous states, enabling anomaly localization, but its effectiveness is constrained by the limited expressiveness of crafted textual descriptions and the domain gap between text and visual features. To address these limitations, AnomalyCLIP Zhou et al. (2023) learns anomaly-specific text prompts from additional anomalous datasets to facilitate unified zero-shot anomaly segmentation, though it still fundamentally relies on text prompts. Recently, InCLTR Zhu & Pang (2024) proposes using normal image samples as in-context prompts, improving anomaly detection performance through collaborative text and image prompting. Despite these advances, all of these methods depend heavily on textual prompts, which restrict their ability to achieve precise segmentation boundaries compared to visual promptable models Kirillov et al. (2023).

Figure 1: An illustration of achieving iCAS via General-to-Specific pretraining.

Promptable visual in-context models such as SAM Kirillov et al. (2023) have demonstrated remarkable generalization capabilities through large-scale pre-training on semantic segmentation datasets, enabling interactive segmentation across a variety of tasks. Nevertheless, SAM's performance in anomaly segmentation is limited, often requiring extensive user interactions to produce accurate results Ji et al. (2024); Yang et al. (2024). This limitation is primarily attributed to the substantial domain gap between SAM's pre-training data and real-world anomaly distributions. As an interactive segmentation model, SAM is not inherently able to perform automated cross-image inference based on in-context images and masks. To address this issue, approaches like PerSAM Cheng et al. (2021) and Matcher Liu et al. (2024c) perform customized in-context inference by feeding SAM with localization priors from extra modules. Also, SegIC Meng et al. (2024) a Matcher Liu et al. (2024c) propose an end-to-end model trained from scratch to perform in-context segmentation with sample prompts. These methods highlight that current visual in-context models struggle to generalize directly to anomaly segmentation and often require the collaboration of multiple auxiliary models.

To address these challenges, we propose the **In-Context Anomaly Segmentation** (iCAS) model together with a **General-to-Specific pre-training** paradigm, as show in Fig. 1. The iCAS model is designed to enhance in-context reasoning and enable precise anomaly localization, leveraging limited anomaly-specific data. Meanwhile, the General-to-Specific strategy aims to systematically overcome the scarcity and diversity limitations of anomalous samples, progressively refining the model's capacity to handle diverse anomaly types. Together, these innovations enable robust and scalable anomaly segmentation without the need for extensive anomaly-specific datasets.

Specifically, iCAS is adapted from the mask classification transformer architecture Cheng et al. (2021) and incorporates an in-context query matching mechanism. The model first generates a set of class-agnostic masks using a mask decoder and subsequently matches and aggregates these masks through an in-context transformer decoder by comparing semantic tokens extracted from the target and reference images. To further enhance feature selection, we introduce a Greedy Query Selection (GQS) mechanism, which applies an active learning strategy to sample representative features from the image embeddings as content queries. During inference, iCAS is capable of flexible in-context reasoning based on various forms of context, including interactive masks, anomaly-mask pairs, or normal reference samples. To increase the model's sensitivity to subtle anomalous content, we propose the Mask-level Feature Matching (MFM) module, which facilitates finer discrimination between normal and abnormal regions.

To fully exploit the potential of iCAS under limited anomaly data conditions, we introduce the General-to-Specific training paradigm. In this paradigm, the model is first pre-trained on large-scale generic semantic segmentation datasets to acquire strong mask prediction and in-context segmentation capabilities. It is subsequently re-trained on anomaly-specific datasets, allowing it to refine its ability to predict anomaly masks using the limited available anomalous samples. This two-stage strategy bridges the gap between generic semantic understanding and precise anomaly localization, effectively leveraging the strengths of both abundant generic data and scarce anomaly-specific examples.

Overall, our contribution can be summarized as:

- We introduce the iCAS model, which utilizes an in-context transformer, a greedy query selection mechanism, and mask-level feature matching to enhance the localization of anomalous regions.

- We propose a General-to-Specific paradigm for in-context anomaly segmentation pre-training, aiming to bridge the gap between general semantic segmentation and specialized anomaly segmentation.

- Extensive experiments and ablation studies are conducted across various anomaly segmentation and detection tasks to validate the effectiveness of the proposed methods.

## 2 RELATED WORK

**Anomaly Segmentation**    Visual anomaly segmentation is usually defined as unsupervised learning that discriminates fine-grained anomalies based on normal samples Pang et al. (2021). Considering a few anomaly samples available in real application scenarios, some recent studies have also focused on open-set supervised anomaly segmentation tasks Ding et al. (2022); Zhu et al. (2024). Anomaly segmentation generally includes reconstruction-based Bergmann et al. (2018); Shi et al. (2021); You et al. (2022); Salehi et al. (2021); Deng & Li (2022); Bergmann et al. (2020), synthesis-based Li et al. (2021); Zavrtanik et al. (2021); Zhang et al. (2024b); Liznerski et al. (2020), and embedded-based Defard et al. (2021); Roth et al. (2022); Xie et al. (2023) methods, while its core proposition is to discover and learn powerful discriminative representations to recognize anomalous patterns. In particular, some studies Santos et al. (2023); Costanzino et al. (2024) have found that robust pre-trained visual representations achieve powerful anomaly segmentation performance without training. To obtain anomaly segmentation models with transferability and generalizability, there are also studies to design specialized visual pre-training paradigms, such as RegAD with a feature registration model Huang et al. (2022) and MetaUAS with a change detection model Gao (2024), to achieve few-shot anomaly detection. Recently, large pre-trained vision-language models, such as CLIP Radford et al. (2021), have demonstrated strong promptable perception and in-context transfer abilities on downstream visual tasks. WinCLIP Jeong et al. (2023) presents to enhance CLIP with handcrafted text prompts to enable powerful zero-shot and few-shot visual anomaly segmentation performance. AnomalyCLIP Zhou et al. (2023) further proposes learnable generic textual prompts to optimize the precision of CLIP for anomaly detection and segmentation. Meanwhile, InCTRL Zhu & Pang (2024) incorporates the visual feature residuals of CLIP to achieve contextual anomaly detection for test samples by utilizing a few normal samples as references. Despite the promising results achieved with text–vision alignment, Musc Li et al. (2024a) reveals that comparison between visual features can lead to superior anomaly localization ability. UniVAD Gu et al. (2025) further shows that leveraging the collaborative cues from multiple foundation models with component-level clustering yields a unified training-free few-shot anomaly detection framework across diverse domains. DictAS Qu et al. (2025) enhances CLIP-based anomaly segmentation by introducing a dictionary-learning mechanism that performs sparse dictionary lookup to robustly identify unseen anomalous patterns. Unlike the previously mentioned approaches that use native or fine-tuned language–vision models, we focus on training a large anomaly segmentation model from scratch with vision-only in-context learning.

**In-Context Segmentation**    Recent developments in in-context segmentation began with the Segment Anything Model (SAM) Kirillov et al. (2023), which demonstrated that a single, promptable segmentation network can generalize to diverse objects without task-specific training. After that, MobileSAM Zhang et al. (2023) distills SAM's heavyweight encoder into a compact architecture that supports real-time inference on edge devices. In parallel, HQ-SAM Ke et al. (2023) targets mask fidelity by integrating a learnable high-quality token into the decoder and fusing it at multiple feature levels to produce crisper boundaries and finer details. To extend SAM's cross-image in-context segmentation capability, one-shot approaches such as Matcher Liu et al. (2024c) guide mask prediction by matching deep features between a reference concept and a target image, and Per-SAM Zhang et al. (2024a) personalizes SAM's outputs through a lightweight target-guided attention and minimal semantic prompts. Novel frameworks such as SINE Liu et al. (2024b) propose a simple in-context learning framework to address task ambiguity in promptable segmentation by training a mask transformer decoder Cheng et al. (2021) from scratch with frozen DINOv2 features. Meanwhile, SegIC Meng et al. (2024) proposes to utilize meta-dense correspondences, such as the visual features of DINOv2 or the vision-language features of CLIP, to train the mask decoder for potential mask prediction. However, recent studies reveal the limitations of SAM for anomaly detection or surface defect segmentation Ji et al. (2024); Cao et al. (2023); Yang et al. (2024), where more interactions and guidance are usually required to obtain more accurate mask prediction boundaries. Thus, we propose a generalized in-context anomaly segmentation model and training strategy to achieve superior discrimination of anomaly boundaries.

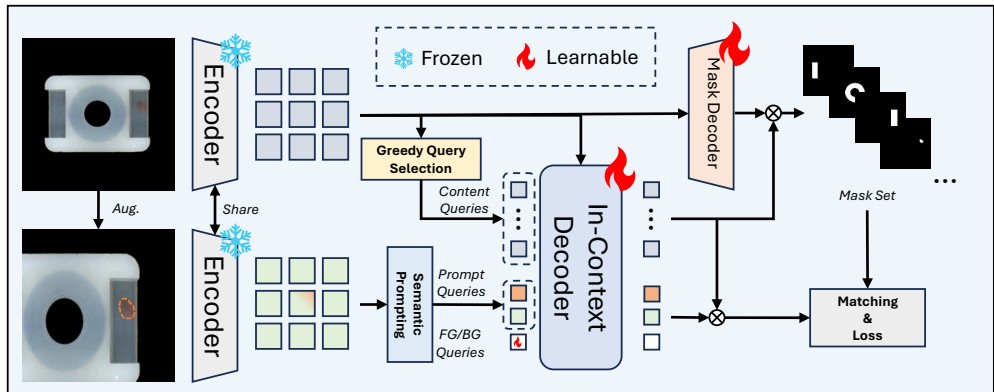

Figure 2: An overview of the iCAS framework. We use a frozen pre-trained encoder to extract visual features, and then train the mask decoder and the in-context decoder. For in-context learning, we use mask pooling to obtain prompt queries and greedy query selection to obtain content queries for context token learning, which are then used for mask set matching.

## 3 METHOD

In this section, we first introduce the proposed visual **In-Context Anomaly Segmentation** (iCAS) framework in Sec.3.1, which consists of a mask decoder that generates a diverse set of candidate masks and an in-context transformer decoder that matches these masks to anomaly prompts. Notably, we propose a novel content query sampling method, Greedy Query Sampling, to realize sampling class-agnostic queries from image features. After that, we present the objective of in-context learning in Sec.3.2. In Sec.3.3, we introduce a General-to-Specific training strategy, which includes generic pre-training and anomaly-aware pre-training. Generic pre-training aims to learn effective potential mask generation and semantic prompt matching on a large-scale semantic segmentation dataset. Considering the fine-grained representation in the anomaly samples, we pre-train the model on extra anomaly detection datasets to refine the perception of anomaly content and boundaries. The inference phase is described in Sec.3.4, iCAS performs in-context anomaly segmentation from different prompt formulations on novel scenarios. Particularly, we introduce semantic reasoning as well as a Mask-level Feature Matching module for enhancement.

### 3.1 UNIFIED FRAMEWORK

Given a collected dataset $D$, we randomly select a sample $\{I_t, M_t\}$ as the target image and its corresponding ground-truth mask. To construct a reference sample for prompting, we apply data augmentation to $\{I_t, M_t\}$, resulting in a new pair $\{I_r, M_r\}$ as reference. The detailed architecture of our proposed iCAS is shown in Fig. 2. We use a pre-trained DINOv2 Oquab et al. (2023) as a feature encoder to extract robust visual features from input images. For each image pair $I_r$ and $I_t$, we extract the $F_r, F_t \in \mathbb{R}^{C \times H_0 \times W_0}$ as the the extracted features with the frozen image encoder, where $C$, $H_0$ and $W_0$ represent the number of feature channels, and the height and width of the feature maps.

**Semantic prompting** aims to extract prompt queries from the in-context examples. Given the reference image $I_r$ and its corresponding promptable mask $M_r$, we apply a *MaskPooling* operation to generate semantic prompt queries. Specifically, for each semantic instance, we aggregate the visual features within the regions specified by $M_r$ and $1 - M_r$, to generate in a set of (abnormal) prompt queries $Q_p^0 \in \mathbb{R}^{(N) \times C}$ and (normal) non-target prompt $Q_{ntp}^0 \in \mathbb{R}^{(N) \times C}$ for $N$ semantic instances.

**In-context transformer** takes a typical transformer decoder structure Carion et al. (2020). The transformer decoder typically uses randomly initialized query embeddings and positional encodings as content queries $Q_c^0 \in \mathbb{R}^{K \times C}$, where $K$ denotes the number of context queries. In our framework, we additionally use prompt queries $Q_p^0, Q_{ntp}^0$ and learnable foreground&background queries $Q_{fg}^0, Q_{bg}^0$ to support in-context anomaly recognition. In each transformer decoder layer, the initial context query set $\{Q_c^0, Q_p^0, Q_{ntp}^0, Q_{fg}^0, Q_{bg}^0\}$ is transformed with the target image feature $F_t$ through multi-

head self- and cross-attention mechanisms. We eventually receive the output query embedding sets $\{Q_c, Q_p, Q_{ntp}, Q_{fg}, Q_{bg}\}$ with global context information after multiple layers of transformer.

**Greedy Query Selection** (GQS) is proposed to sample representative embeddings as content queries, which is inspired by conditional query selection to promote the convergence of transformer Zhu et al. (2020); Zhang et al. (2022b); Li et al. (2023); Liu et al. (2024a). Unlike these methods, which require a category prior as a condition, GQS covers representative features and preserves outliers through active learning. To be specific, we convert the GQS into a k-center

---

**Algorithm 1** Greedy Query Selection

**Input:** target features $F_t \in \mathbb{R}^{H_0 W_0 \times C}$
Initialize $\mathbf{Q_c^0} = \{\ \}$
**for** $i \in (0, K - 1)$ **do**
$\quad q = \arg\max_{i \in F_t - Q_c^0} \min_{j \in F_t} <i, j>$
$\quad \mathbf{Q_c^0} = \mathbf{Q_c^0} \cup \{q\}$
**end for**
**Return $\mathbf{Q_c^0}$**

---

problem Farahani & Hekmatfar (2009); Sener & Savarese (2017), i.e., select $K$ centroid features such that the maximum distance between each image feature and its nearest centroid feature is minimized. The GQS is illustrated in Algorithm 1, and more details are given in the Appendix.

**Mask decoder** takes the target image features $F_t$ as input and outputs the pixel embeddings $P_t \in \mathbb{R}^{C \times H \times W}$ gradually via upsampling layers. Inspired by the mask classification modelCheng et al. (2021), we compute the dot product of content query embeddings $Q_c$ and pixel embeddings$P_t$ to generate class-agnostic mask sets $\mathcal{M}_t \in \mathbb{R}^{K \times H \times W}$, i.e. $\mathcal{M}_t = sigmoid(Q_c \times P_t)$. After pre-training on a large-scale dataset, the mask prediction has stronger transferability compared to the per-pixel predictionZhang et al. (2022a), providing iCAS with powerful in-context semantic segmentation ability.

## 3.2 OBJECTIVE

We predict the $N$ class probabilities for content queries $Q_c$ using $\{Q_p, Q_{ntp}, Q_{bg}\}$ and $\{Q_{fg}, Q_{ntp}, Q_{bg}\}$ as semantic classifiers and non-target classifiers to obtain $z = \{(p_p^i, p_{ntp}^i, \mathcal{M}_t^i)\}_{i=1}^K$ probability-mask pairs. Given $K$ predicted probability-mask pairs $z$ and $K_t$ ground-truth segments $z_t = \{(c_t^i, M_t^i)\}_{i=1}^{K_t}$, we compute a matching cost, i.e., $-p_p^i(c_t^j) - p_{ntp}^i(c_t^j) + \mathcal{L}_{mask}(\mathcal{M}_t^i, M_t^i)$, to assign predictions to ground truths via the bipartite matching. $p_p^i(c_t^j)$ and $p_{ntp}^i(c_t^j)$ are target prediction probabilities and non-target prediction classifiers, respectively, where $p_{ntp}^i(c_t^j)$ is designed to recognize unknown content based on known prompts. $\mathcal{L}_{mask}$ is a binary mask loss followingZhang et al. (2022a). For the $j$th ground truth, we get a $\sigma(j)$ index from the optimal assignment of bipartite matching.

To train the parameters of iCAS, the overall in-context learning loss is defined as:

$$L_{ic}(z, z_t) = \sum_{j=1}^K [-\log p_p^\sigma(j)(c_t^j) - \log p_{ntp}^\sigma(j)(c_t^j) + \mathcal{L}_{mask}(\mathcal{M}_t^{\sigma(j)}, M_t^i)], \tag{1}$$

which includes a prompt cross-entropy loss, a non-target prompt cross-entropy loss and a mask loss. Notably, prompt classification and mask prediction are derived from maskformer Cheng et al. (2021), while the non-target prompt optimization aims to enhance the segmentation of abnormal objects.

## 3.3 GENERIC-TO-SPECIFIC TRAINING

We propose a **General-to-Specific** training strategy to enable iCAS to achieve accurate in-context anomaly segmentation: a generic pre-training phase followed by an anomaly-specific pre-training phase. First, in the **generic pre-training phase**, iCAS is trained on large-scale semantic segmentation datasets, denoted as $\mathcal{D}_g$. Each training sample in $\mathcal{D}_g$ contains various objects and their corresponding masks, enabling iCAS to learn to generate a set of class-agnostic potential masks that decompose an image into multiple segments with precise semantic boundaries. Next, in the **anomaly-specific training phase**, we further train iCAS on a semantic anomaly dataset $\mathcal{D}_s$, where each anomalous region is annotated with a semantic mask. For each anomaly sample, we provide a single anomaly prompt and target, and optimize the predicted mask set to best fit the fine-grained anomaly region. This phase specifically optimizes iCAS for accurately predicting the anomaly boundaries within the in-context learning framework.

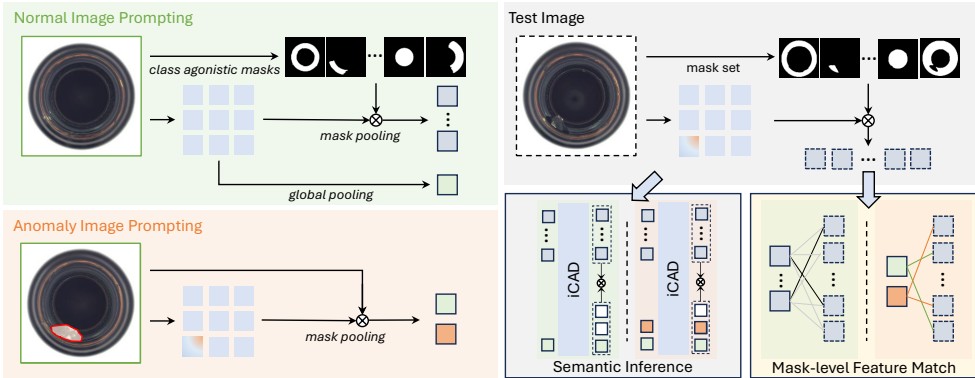

Figure 3: An overview of the in-context inference for iCAS. We present the prompting process for normal and anomaly images, respectively. Then, we show the inference process of iCAS based on normal and anomaly prompts. Specifically, in semantic inference and mask-level feature matching, the left and right sides refer to in-context matching for normal and anomaly prompts, respectively.

With the general-to-specific pre-training paradigm, iCAS can learn to achieve generalization from the generic data in $\mathcal{D}_g$ and improved sensitivity for anomaly segmentation through fine-tuning on $\mathcal{D}_s$. During inference, iCAS is applied to the unseen anomaly dataset, denoted as $\mathcal{T}$, which represents the target dataset for in-context reasoning. Importantly, $\mathcal{T}$ and $\mathcal{D}_s$ are disjoint datasets, meaning they do not share any samples. This ensures that $\mathcal{T}$ represents a completely separate domain from $\mathcal{D}_s$, with no overlap between the two. The model generates predictions for each test sample from $\mathcal{T}$, where the input is an image and its corresponding anomaly prompt, producing the predicted mask.

Previous methods in anomaly segmentation typically rely on anomaly-specific data augmentation applied to normal samples in the target dataset Li et al. (2021), or generate synthetic anomalies for training on the target dataset Zavrtanik et al. (2021); Zhang et al. (2024b). Other approaches utilize meta-learning techniques on generic datasets to achieve domain adaptation Gao (2024); Wu et al. (2021). In contrast, our method follows a General-to-Specific paradigm, where iCAS is pre-trained on a large-scale generic dataset and fine-tuned on an anomaly-specific dataset. This allows iCAS to generalize across domains while specializing in anomaly segmentation, without the need for direct anomaly synthesis or domain-specific training. Our approach is designed for in-context anomaly segmentation, enabling iCAS to adapt to a wide range of anomaly segmentation tasks and ensuring robust performance even in novel domains.

## 3.4 IN-CONTEXT ANOMALY SEGMENTATION

We demonstrate the in-context inference phrase in Fig. 3, which is described in detail as follows.

**Semantic Inference.** The proposed iCAS can utilize anomaly image prompting and normal image prompting for in-context anomaly segmentation. Given $i$th anomaly sample and its prompt mask pair for anomaly image prompting, we assign a pixel-level semantic scores at position $[h, w]$ via semantic inference Zhang et al. (2022a), i.e. $\mathcal{S}_s = \sum_{i=1}^{K} p^i \cdot \mathcal{M}_t^i[h, w]$, where $p^i = \frac{\exp(<Q_p, Q_c^i>)}{\sum_{q \in \{Q_p, Q_{ntp}, Q_{bg}\}} \exp(<p, Q_c^i>)}$. Intuitively, $\mathcal{S}_s$ is only used as a result of semantic segmentation of known anomalous samples, which is consistent with few-shot or interactive prompt segmentation Kirillov et al. (2023); Zhang et al. (2024a); Liu et al. (2024c;b). For normal image prompting, we propose a normal promoting anomaly score $\mathcal{S}_n = \sum_{i=1}^{K} p^i \cdot \mathcal{M}_t^i[h, w]$, where $p^i = \frac{\exp(<Q_{fg}, Q_c^i>)}{\sum_{q \in \{Q_{ntp}, Q_{fg}, Q_{bg}\}} \exp(<p, Q_c^i>)}$. The proposed $\mathcal{S}_n$ for anomaly detection is motivated by a negative likelihood function Rai et al. (2023) that estimates non-target content, i.e., normal prompts, from the predicted queries and masks.

**Mask-level Feature Matching.** We further introduce the mask-level feature matching (MFM), where feature-level comparison is performed for identifying anomalous regions with the potential mask set. Given the feature map $F_t$ and the set of potential masks $\mathcal{M}_t$ of a target sample, we calculate their mask-level embeddings $Q_m$ through the *MaskPooling* operation. The anomaly image prompting use

anomaly embedding as semantic prompts $Q_p^0$ and normal embedding as non-targeted prompts $Q_{ntp}^0$, the MFM score is computed as $\mathcal{S}_s^{mfm} = \sum_{i=1}^{K} p^i \cdot \mathcal{M}_t^i[h,w]$, where $p^i = \frac{\exp(<Q_p^0, Q_m^i>)}{\sum_{q \in \{Q_p^0, Q_{ntp}^0\}} \exp(<p, Q_m^i>)}$. Furthermore, for normal image prompting with a reference $I_r$, we first get its potential masks $M_r$ by mask decoder and then compute the mask-level prompt embedding $Q_r \in \mathbb{R}^{K \times C}$ set by mask pooling. The MFM score with normal prompts is compute as $\mathcal{S}_n^{mfm} = \sum_{i=1}^{K} p^i \cdot \mathcal{M}_t^i[h,w]$, where $p^i = \min_{j \in K} 1- <Q_m^i, Q_r^j>$. Note that we add the MFM score as a complement to the semantic inference score to strengthen robustness, i.e. $\mathcal{S}_{s,n} = \mathcal{S}_{s,n} + \mathcal{S}_{s,n}^{mfm}$.

## 4 EXPERIMENTS

### 4.1 DATASETS

**Training Data.** For generic pre-training, we build a large-scale image dataset with semantic annotations, collected from COCO Lin et al. (2014), ADE20K Zhou et al. (2017), and Objects365 Shao et al. (2019), and we use SAM Kirillov et al. (2023) to generate additional masks for categories lacking semantic annotations. To achieve generalizability on the AD task, we collect annotated abnormal samples from large-scale AD datasets, RealIAD Wang et al. (2024) and MANTA Fan et al. (2024) for anomaly-aware pre-training. The collected anomaly pre-training dataset consists of 85,564 images in over 300 anomaly categories.

**Evaluation Data.** We evaluate our proposed method on multiple anomaly detection benchmarks. For the semantic anomaly segmentation tasks, we conduct few-shot segmentation and interactive segmentation experiments on the MVTecAD Bergmann et al. (2019) with 15 categories and the VisA Zou et al. (2022) dataset with 12 categories. The detailed categories are listed in the Appendix A.2. For the few-shot AD and open-set AD tasks, we additionally use the SDD Tabernik et al. (2020), ELPV Deitsch et al. (2019), and AFID Silvestre-Blanes et al. (2019) datasets as benchmarks.

### 4.2 IMPLEMENTATION DETAILS

For both generic pre-training and anomaly-aware re-training, we use the same training setup despite there are different objectives. We use the pre-trained DINOv2 as the feature encoder and keep its parameters frozen, and train the parameters of the mask decoder, in-context transformer, and learnable tokens. The learning rate is set to $1e^{-4}$ and the batch size is set to 32. For both stages, we use AdamW with a weight decay of $5e^{-2}$ to optimize the model. We set the image size to $518 \times 518$ and use a strong data augmentation strategy Ghiasi et al. (2021) to generate image-prompt pairs. Note that generic pre-training takes 50 epochs and anomaly-aware re-training takes 5 epochs. For evaluation, we use mIoU for semantic segmentation and AUROC for anomaly detection.

### 4.3 MAIN RESULTS

Table 1: Semantic Anomaly Segmentation Results on MVTecAD and VisA in terms of mIoU.

| MVTecAD | CP | BT | HN | LT | CB | CS | GD | PL | TS | MN | SR | TB | ZP | TL | WD | Mean |
|---|---|---|---|---|---|---|---|---|---|---|---|---|---|---|---|---|
| PerSAM | 5.4 | 16.3 | 7.4 | 3.5 | 25.9 | 5.2 | 3.6 | 9.0 | 24.8 | 27.6 | 8.9 | 24.8 | 27.6 | 33.0 | 4.1 | 15.1 |
| Matcher | 1.6 | 16.1 | 4.8 | 0.7 | 12.2 | 1.4 | 0.8 | 1.6 | 16.7 | 2.5 | 3.0 | 7.0 | 3.2 | 2.7 | 4.5 | 5.3 |
| SINE | 19.5 | 17.4 | 6.0 | 28.1 | 18.0 | 13.7 | 8.9 | 33.6 | 21.9 | 4.3 | 51.3 | 8.1 | 34.9 | 11.5 | 10.3 | 19.2 |
| Ours | **54.6** | **24.9** | **35.5** | **66.7** | **44.1** | **78.2** | **61.2** | **84.7** | **78.0** | **27.6** | **63.6** | **24.8** | **51.6** | **59.0** | **29.5** | **51.1** |

| VisA | CD | CP | CS | CG | FR | M1 | M2 | P1 | P2 | P3 | P4 | PF | Mean |
|---|---|---|---|---|---|---|---|---|---|---|---|---|---|
| PerSAM | 1.7 | 5.5 | 10.3 | 1.3 | 15.4 | 0.2 | 0.1 | 3.3 | 1.1 | 0.5 | 6.4 | 13.0 | 4.9 |
| Matcher | 0.9 | 7.9 | 2.4 | 0.4 | 0.1 | 0.3 | 0.2 | 2.7 | 1.0 | 7.6 | 4.3 | 4.6 | 2.7 |
| SINE | 1.2 | 3.1 | 11.9 | 3.9 | 20.5 | 0.5 | 0.4 | 17.2 | 7.3 | 7.4 | 16.8 | 17.4 | 8.9 |
| Ours | **25.9** | **58.3** | **75.5** | **65.5** | **42.0** | **2.4** | **2.2** | **27.6** | **22.9** | **15.2** | **27.8** | **82.5** | **37.3** |

**Semantic Anomaly Segmentation.** The evaluation of semantic anomaly segmentation focuses on comparisons with PerSAM Zhang et al. (2024a), Matcher Liu et al. (2024c), and SINE Liu et al. (2024b), methods that aim to perform contextual segmentation from a prompt example without training. In particular, we take one sample from each of all the anomaly types from each object category as prompts, and then evaluate them on each category individually. The overall results on the MVTecAD and VisA datasets are shown on Fig. 1, where each object category is denoted by an abbreviation. The results demonstrate that current general in-context segmentation models fail on anomaly segmentation tasks, despite their impressive performance on general semantic segmentation tasks. Notably, our proposed iCAS model achieves surprising generalization and robust performance on semantic anomaly segmentation, achieving an average mIoU of 51.1% and 37.3% on MVTecAD and VisA datasets, respectively. Compared to the state-of-the-art (SOTA) method, SINE Liu et al. (2024b), our method achieves a remarkable increase of 31.9% mIoU on MVTecAD and 28.4% mIoU on VisA. Our results show that iCAS exhibits powerful training-free anomaly segmentation ability, which can directly generate accurate segmentation masks in a wide range of anomaly detection scenarios.

Table 2: Evaluation of Interactive Anomaly Segmentation on MVTecAD and VisA in terms of mIoU.

| MVTecAD | CP | BT | HN | LT | CB | CS | GD | PL | TS | MN | SR | TB | ZP | TL | WD | Mean |
|---|---|---|---|---|---|---|---|---|---|---|---|---|---|---|---|---|
| SAM | 42.5 | 41.6 | 50.0 | 49.8 | 51.1 | 31.5 | 31.7 | 39.8 | 32.4 | 44.1 | 33.0 | 22.4 | 14.5 | 63.2 | 27.5 | 39.1 |
| MB-SAM | 17.2 | 34.7 | 48.6 | 28.5 | 48.9 | 31.2 | 8.8 | 32.4 | 30.9 | 38.0 | 34.4 | 16.3 | 15.4 | 66.7 | 17.7 | 32.1 |
| HQ-SAM | 53.1 | 44.4 | 51.2 | 59.8 | **54.4** | 37.8 | 38.7 | 55.4 | 41.5 | **63.9** | 40.1 | 39.0 | 19.6 | **80.1** | 49.9 | 48.8 |
| Ours | **74.8** | **44.6** | **51.6** | **72.2** | 52.4 | **81.9** | **69.6** | **86.8** | **81.1** | 35.3 | **79.5** | **48.4** | **55.3** | 69.7 | **52.4** | **63.7** |

| VisA | CD | CP | CS | CG | FR | M1 | M2 | P1 | P2 | P3 | P4 | PF | Mean |
|---|---|---|---|---|---|---|---|---|---|---|---|---|---|
| SAM | 2.5 | 22.2 | 3.1 | 28.8 | 4.6 | 4.3 | 2.7 | 10.2 | 11.5 | 20.0 | 2.3 | 9.5 | 10.7 |
| MB-SAM | 2.7 | 5.8 | 1.0 | 31.0 | 3.4 | 2.8 | 2.8 | 8.8 | 8.9 | 16.4 | 2.1 | 11.1 | 8.1 |
| HQ-SAM | 9.0 | 26.7 | 8.5 | 35.9 | 5.9 | 6.5 | 3.5 | 11.5 | 12.4 | 19.2 | 2.3 | 15.2 | 13.1 |
| Ours | **35.7** | **67.4** | **71.6** | **67.8** | **68.1** | **7.0** | **4.7** | **47.3** | **30.4** | **32.4** | **39.4** | **83.2** | **46.3** |

**Interactive Anomaly Segmentation.** The evaluation of interactive anomaly segmentation follows promptable segmentation task proposed in SAM Kirillov et al. (2023). As show in Tab. 2, we compare the iCAS with state-of-arts interactive segmentation model, including SAM Kirillov et al. (2023), Mobile-SAM Zhang et al. (2023) and HQ-SAM Ke et al. (2023). For these methods, we directly input the point coordinates to obtain interactive segmentation results. Particularly, we use a simplified interactive approach for iCAS, which is to convert points or boxes into area masks as prompts. Note that we randomly sample positive and negative point pairs for all methods as the interactive prompts. Evidently, our proposed iCAS outperforms the competing methods by a significant margin. Furthermore, while these interactive models still have impressive results on MVTecAD, they fail to segment tiny defects on VisA, of which our approach has a notable improvement.

Table 3: Comparison of few-shot anomaly detection AUROC on MVTecAD (MVT) Bergmann et al. (2019), VisA Zou et al. (2022), SDD Tabernik et al. (2020), ELPV Deitsch et al. (2019), and AFID Silvestre-Blanes et al. (2019) datasets. The results of the other methods are quoted from Zhu & Pang (2024); Li et al. (2024b), while ours is the mean of 5 random runs.

| Method | 2-shot | | | | | 4-shot | | | | | 8-shot | | | | |
|---|---|---|---|---|---|---|---|---|---|---|---|---|---|---|---|
| | MVT | VisA | SDD | ELPV | AFID | MVT | VisA | SDD | ELPV | AFID | MVT | VisA | SDD | ELPV | AFID |
| RegAD | 82.4 | 55.7 | 49.9 | 57.1 | 56.4 | 85.7 | 57.4 | 52.5 | 59.6 | 59.6 | 88.2 | 58.9 | 59.4 | 63.3 | 60.3 |
| PatchCore | 85.8 | 81.7 | 90.2 | 71.6 | 73.9 | 88.5 | 84.3 | 92.3 | 75.6 | 73.3 | 92.2 | 86.0 | 92.5 | 83.7 | 74.5 |
| WinCLIP | 93.1 | 84.2 | 94.2 | 72.6 | 72.6 | 94.0 | 85.8 | 94.3 | 75.4 | 76.4 | 94.7 | 86.8 | 94.1 | 81.4 | 79.6 |
| InCTRL | 94.0 | 85.8 | 97.2 | 83.9 | 76.1 | 94.5 | 87.7 | 97.5 | 84.6 | 79.0 | 95.3 | 88.7 | 97.8 | 87.2 | 80.6 |
| One2N | 95.1 | 87.2 | 96.8 | **85.6** | 78.3 | 95.6 | 88.6 | 97.8 | **87.3** | 82.6 | 96.2 | 89.9 | 98.4 | **90.6** | 84.7 |
| Ours | **97.6** | **92.8** | **98.4** | 84.2 | **81.2** | **97.8** | **93.0** | **98.8** | 84.0 | **88.8** | **98.2** | **93.9** | **99.4** | 88.0 | **88.7** |

**Few-shot Unsupervised Anomaly Detection.** Our proposed iCAS can directly perform unsupervised anomaly detection tasks that rely only on normal samples. However, other visual in-context

models are unable to utilise only non-target prompts, apart from utilizing additional AD model guidance Cao et al. (2023); Li et al. (2025). We compare iCAS with SOTA few-shot AD models and the results are presented in Tab. 3. It can be noticeably observed that the performance of iCAS is more powerful than other few-shot AD algorithms on various datasets. We also show the results with increasing number of shots to verify the robustness and scalability of iCAS. The overall experiments prove that our proposed iCAS has a strong generalization ability with anomaly detection tasks.

**Open-set Supervised Anomaly Detection.** Inspired by open-set supervised anomaly detection studies Ding et al. (2022), we allow iCAS to detect anomalies using normal samples with a few annotated anomalous samples. We compare with the SOTA methods DevNet Pang et al. (2019), DRA Ding et al. (2022) and AHL Zhu et al. (2024) and follow their settings to randomly select an anomalous sample as input. This task involves exploiting the open

Table 4: Open-set anomaly detection results in terms of AUROC

| Method | MVTecAD | SDD | ELPV | AFID |
|--------|---------|------|------|------|
| DevNet | 83.2 | 85.1 | 81.0 | 60.9 |
| DRA | 88.9 | 90.7 | 67.6 | 69.3 |
| AHL | 90.1 | 90.9 | 72.3 | 73.4 |
| Ours | **99.2** | **98.5** | **88.1** | **91.9** |

set recognition ability of the AD models in scenarios where anomalous samples are available. The overall results are shown in Tab. 4, where our method significantly outperforms the other methods.

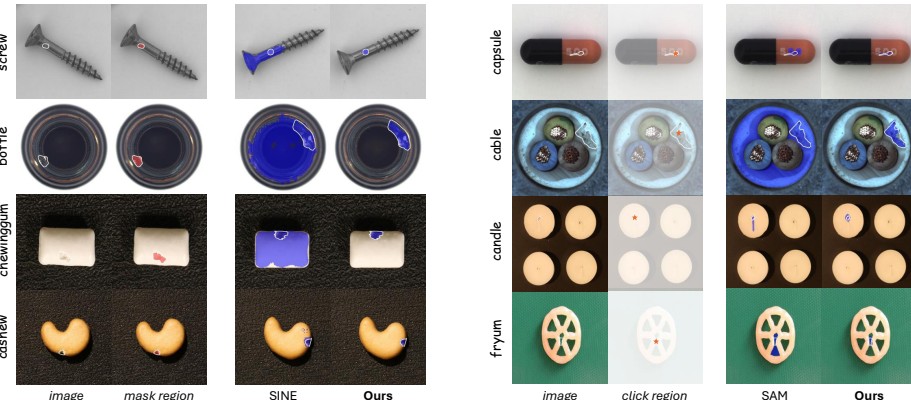

Figure 4: Qualitative comparison of in-context anomaly segmentation on MVTecAD and VisA.

**Evaluation on Medical Datasets.** To demonstrate the effectiveness of our method in the medical domain, we conducted additional experiments on Retinal OCT Hu et al. (2019); Kermany et al. (2018), Brain MRI Baid et al. (2021), and Liver CT Landman et al. (2015); Bilic et al. (2023) datasets. The results in Tab. 5&6 show that iCAS significantly outperforms existing in-context segmentation models in both interactive and semantic settings. Specifically, in interactive anomaly segmentation, iCAS consistently surpasses SAM, notably more than doubling the mIoU on Brain MRI and Liver CT. Furthermore, in semantic anomaly segmentation, iCAS demonstrates superior robustness compared to PerSAM and Matcher, particularly on the Liver CT dataset where it achieves a substantial lead. These confirm that iCAS possesses strong generalization capabilities, extending its efficacy effectively from industrial inspection to complex medical imaging tasks.

Table 5: Results of interactive anomaly segmentation of SAM and iCAS on medical datasets

| Dataset | Method | mIoU | FB-IoU |
|---------|--------|------|--------|
| Retinal OCT | SAM | 43.5 | 63.7 |
| | iCAS | 54.5 | 72.0 |
| Brain MRI | SAM | 21.0 | 52.7 |
| | iCAS | 49.4 | 72.9 |
| Liver CT | SAM | 20.5 | 58.4 |
| | iCAS | 42.3 | 70.6 |

Table 6: Results of semantic anomaly segmentation on medical datasets

| Dataset | Method | mIoU | FB-IoU |
|---------|--------|------|--------|
| Retinal OCT | PerSAM | 13.6 | 41.7 |
| | Matcher | 8.3 | 34.2 |
| | iCAS | 31.0 | 61.9 |
| Brain MRI | PerSAM | 13.6 | 53.6 |
| | Matcher | 10.7 | 42.9 |
| | iCAS | 15.1 | 55.2 |
| Liver CT | PerSAM | 14.1 | 15.8 |
| | Matcher | 6.5 | 11.4 |
| | iCAS | 47.3 | 66.8 |

## 4.4 VISUALIZATION

To qualitatively evaluate the effectiveness of our method, we show comparison of the context segmentation results with other SOTA models in Fig. 4. In semantic anomaly segmentation, our method can precisely segment anomalous regions using prompts of masked regions, while the generic model SINE Liu et al. (2024b) is insensitive to anomaly boundaries. In addition, our method allows for segmenting anomalous regions by interactive prompting, e.g., clicking on the region. Although SAM Kirillov et al. (2023) can generate precise boundaries, it suffers from semantic ambiguity in segmenting unexpected objects.

## 4.5 ABLATION STUDY

Table 7: Ablations of the training strategy and proposed modules, GQS and MFM. G-Train and A-Train indicate the generic pre-training and anomaly-aware pre-training.

| G-Train | A-Train | GQS | MFM | MVTecAD | VisA |
|---|---|---|---|---|---|
| ✓ | | | | 20.0 | 11.0 |
| | ✓ | | | 20.9 | 21.9 |
| ✓ | ✓ | | | 47.2 | 32.9 |
| ✓ | | | | 20.0 | 11.0 |
| ✓ | | ✓ | ✓ | 29.2 | 13.6 |
| ✓ | ✓ | | | 47.2 | 32.9 |
| ✓ | ✓ | ✓ | | 49.1 | 35.6 |
| ✓ | ✓ | | ✓ | 48.9 | 35.3 |
| ✓ | ✓ | ✓ | ✓ | **51.1** | **37.3** |

As shown in Tab. 7, we conduct experiments on different combinations of components for iCAS. First, we train the models on the general semantic training set and the anomaly training set individually and evaluate the results separately, and our proposed general-to-specific anomaly training strategy significantly outperforms both the general pre-training and anomaly-aware pre-training. In particular, iCAS learns to generate category-agnostic mask sets from a large-scale general semantic dataset, while the anomaly-specific dataset can optimize the boundaries of the anomaly masks matched for iCAS. Secondly, we conduct ablation studies for the proposed greedy query sampling strategy and mask-level feature matching module. For both generic pre-trained models and fully pre-trained models, our proposed GQS and MFM are able to significantly improve their anomaly segmentation performance, which demonstrates the adaptability of our proposed methods. Specifically, GQS and MFM can improve the mIoU of generic models by $9.2\%$ and $2.6\%$ and can further improve the mIoU of iCAS by $3.9\%$ and $4.4\%$ on MVTecAD and VisA, respectively.

Table 8: Ablation studies according to dataset size and model capacity.

| Datasets | MVTecAD | | VisA | |
|---|---|---|---|---|
| | mIoU | FB-IoU | mIoU | FB-IoU |
| RealIAD | 43.3 | 68.4 | 20.8 | 59.4 |
| MANTA | 41.7 | 67.0 | 30.0 | 59.5 |
| Both | **51.1** | **74.9** | **37.3** | **68.5** |

| Backbone | MVTecAD | | VisA | |
|---|---|---|---|---|
| | mIoU | FB-IoU | mIoU | FB-IoU |
| DINOv2-S | 42.4 | 73.2 | 24.9 | 60.8 |
| DINOv2-B | 45.0 | 73.7 | 33.4 | 68.2 |
| DINOv2-L | **51.1** | **74.9** | **37.3** | **68.5** |

In addition, considering the training cost of iCAS as a large visual model, we perform dataset and model scaling law experiments separately, as shown in Tab. 8. We train the models individually on each of the two large anomaly sample datasets, MANTA and RealIAD, and compare them with the full model trained on them together. In this case, the model trained on RealIAD performs better on MVTecAD, while training on MANTA achieves better results on VisA. Obviously, the full training achieves a significantly best performance. This result reveals that more samples and anomaly types as a training set can enhance the robustness and generalization of the model. Subsequently, we conduct experiments on different backbones, which are small, base, and large models of DINOv2. It can be seen that the performance of the model strengthens as the capacity of the backbone network increases, with larger models leading to stronger feature discrimination and generalization.

## 5 CONCLUSION

We introduce iCAS, an in-context anomaly segmentation model that achieves semantic and interactive prompting anomaly segmentation, as well as normal prompting anomaly detection. Specifically, iCAS is built on an in-context mask transformer and collaborates with our proposed greedy query selection strategy and mask-level feature matching module to capture fine-grained anomaly features. Furthermore, we propose the General-to-Specific pre-training method, which utilizes massive semantic datasets to address the challenge of insufficient pre-training generalizability caused by the scarcity of anomaly datasets. Extensive experiments and ablation studies validate the effectiveness of iCAS.

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

# A    APPENDIX

## A.1    MODEL ARCHITECTURE

For the image encoder, we maintain the structure of DINOv2 Oquab et al. (2023) and freeze the parameters as a robust feature extractor. In addition, we employ the 6 transformer layers and 200 content queries of DETR Carion et al. (2020) as the context transformer decoder, and use the mask decoder structure of maskformer Cheng et al. (2021). The foreground and background tokens are randomly initialized and aligned with the dimensions of the query, and are used to the in-context contrastive learning of prompt training.

## A.2    EXPERIMENT SETTING

We follow the official code implementation of PerSAM Zhang et al. (2024a), Matcher Liu et al. (2024c), and SINE Liu et al. (2024b) for semantic anomaly segmentation and take the same visual prompt samples for fair comparison. For interactive anomaly segmentation, we implement SAM Kirillov et al. (2023), Mobile-SAM Zhang et al. (2023), and HQ-SAM Ke et al. (2023) with the same stochastic sampling points on all experiments.

We use abbreviations for categories in the semantic anomaly segmentation and interactive anomaly segmentation experiments. In detail, MVTecAD is consist of carpet (CP), bottle (BT), hazelnut (HN), leather (LT), cable (CB), capsule (CS), grid (GD), pill (PL), transistor (TS), metalnut (MN), screw (SR), toothbrush (TB), zipper (ZP), tile (TL), wood (WD), while VisA contains candle (CD), capsules (CP), cashew (CS), chewinggum (CG), fryum (FR), macaroni1 (M1), macaroni2 (M2), pcb1 (P1), pcb2 (P2), pcb3 (P3), pcb4 (P4), pipefryum (PF).

## A.3    ANOMALY SEGMENTATION WITH NORMAL AND ANOMALY PROMPTS

We compare the results of normal and anomaly prompting on the results of anomaly segmentation, as shown in Tab. 9. Obviously, the semantic prior of the anomaly prompt produces better results. Moreover, by combining normal and anomaly prompts, more accurate results can be predicted. This is because anomalies are usually open-ended, and even semantic anomalies of the same type exist with unknown semantic features. With the addition of anomaly segmentation with normal sample prompts, better open-set anomaly segmentation results can be achieved.

Table 9: The mIoU results of anomaly segmentation for normal and anomaly prompting.

| MVTecAD | | | VisA | | |
|---|---|---|---|---|---|
| Normal Prompting | Anomaly Prompting | Both | Normal Prompting | Anomaly Prompting | Both |
| 31.7 | 51.1 | 53.2 | 20.9 | 37.3 | 43.4 |

## A.4    MORE RESULTS

To provide a more comprehensive comparison of the in-context model's performance in anomaly segmentation, we evaluate not only the IOU of the anomalous semantic regions but also the IOU metrics for both the foreground and background (FB-IOU). In anomaly segmentation scenarios, the anomalous regions are typically small while the normal regions are much larger, making it important to consider the overall IoU across both foreground and background. As demonstrated in Tab 10, although various models are compared, our proposed iCAS significantly outperforms others, exhibiting exceptional capability in distinguishing both anomalous and normal regions.

As shown in Table 11, DINOv2-Large consistently outperforms both DINOv1-Large and CLIP-ViT-Large across all metrics and datasets. Compared with DINOv1, DINOv2 benefits from larger-scale visual pre-training and improved contrastive/self-supervised objectives, which significantly strengthen its local semantic representation capability. This allows DINOv2 to better capture subtle and small-scale anomalies that are common in industrial inspection scenarios. In contrast, CLIP-ViT-Large mainly emphasizes global image–text alignment during pre-training, leading to weaker local visual discrimination ability and consequently inferior anomaly segmentation performance.

Table 10: The comparison of IoU and FB-IoU results for semantic anomaly segmentation and interactive anomaly segmentation.

| Dataset | MVTecAD | | VisA | |
|---|---|---|---|---|
| Metric | IoU | FB-IoU | IoU | FB-IoU |
| PerSAM | 15.1 | 39.6 | 4.9 | 45.9 |
| Matcher | 5.3 | 23.1 | 2.7 | 32.9 |
| SINE | 19.2 | 52.7 | 8.9 | 48.8 |
| iCAS(Ours) | 51.1 | 74.9 | 37.3 | 68.5 |

| Dataset | MVTecAD | | VisA | |
|---|---|---|---|---|
| Metric | IoU | FB-IoU | IoU | FB-IoU |
| SAM | 39.1 | 48.2 | 10.7 | 45.9 |
| MB-SAM | 32.1 | 38.2 | 8.1 | 45.7 |
| HQ-SAM | 48.8 | 64.7 | 13.1 | 49.7 |
| iCAS(Ours) | 63.7 | 83.8 | 46.3 | 77.0 |

Table 11: Comparison of different backbone models on MVTecAD and VisA datasets.

| Backbone Model | MVTecAD | | VisA | |
|---|---|---|---|---|
| | mIoU (%) | FB-IoU (%) | mIoU (%) | FB-IoU (%) |
| CLIP-ViT-Large | 40.86 | 68.24 | 31.21 | 62.18 |
| DINOv1-Large | 45.40 | 72.92 | 33.87 | 65.43 |
| DINOv2-Large (Ours) | **51.10** | **74.90** | **37.30** | **68.50** |

Table 12: Performance of iCAS under different numbers of queries ($K$) for Interactive and Semantic settings.

| Setting | $K$ Queries | mIoU (%) | FB-IoU (%) |
|---|---|---|---|
| Interactive | 50 | 59.2 | 81.2 |
| | 100 | 59.7 | 81.7 |
| | 150 | 61.3 | 82.2 |
| | 200 | 63.7 | 83.8 |
| Semantic | 50 | 47.2 | 73.8 |
| | 100 | 50.3 | 74.1 |
| | 150 | 50.9 | 74.3 |
| | 200 | 51.1 | 74.9 |

Table 12 reports the performance of iCAS under varying numbers of queries ($K$) for both Interactive and Semantic settings. The results show that the model remains highly stable across different $K$ values. Notably, even when $K$ is reduced to 50, the Interactive mIoU only drops slightly, demonstrating that the Greedy Query Selection (GQS) effectively captures representative features early in the sampling process and is not overly sensitive to the specific choice of $K$.

## A.5 COMPARISON OF ICAS WITH GENERIC ICL MODELS

Table 13: Comparison of iCAS capabilities against generic ICL models. Note that existing models generally lack support for anomaly detection tasks.

| Method | Base Architecture | Training Paradigm | SAS | IAS | FSAD | OSAD |
|---|---|---|---|---|---|---|
| SAM | ViT-H | SA-1B (11,000K) | ✗ | ✓ | ✗ | ✗ |
| SegGPT | Painter (ViT-L) | Painter+ADE20K+COCO…(162K+273K) | ✓ | ✗ | ✗ | ✗ |
| HQ-SAM | SAM (ViT-H) | SA-1B+HQSeg-44K (11,000K+44K) | ✗ | ✓ | ✗ | ✗ |
| Matcher | SAM (ViT-H), DINOv2 (ViT-L) | SA-1B+N/A (11,000K) | ✗ | ✗ | ✗ | ✗ |
| SINE | DINOv2 (ViT-L) | ADE20K+COCO+Obj365 (776K) | ✓ | ✓ | ✗ | ✗ |
| **iCAS (Ours)** | DINOv2 (ViT-L) | **General-to-Specific (861K)** | ✓ | ✓ | ✓ | ✓ |

As shown in Tab. 13, prior approaches face significant challenges in standard anomaly detection settings. Whether in unsupervised few-shot anomaly detection (FSOD) or few-shot anomaly detection with anomalies (Open-set Anomaly Detection, OSAD), existing generic models fail to perform

well. This is because these models are primarily designed to learn semantic alignment without distinguishing between normal and abnormal variations.

On the other hand, iCAS is specifically trained to bridge this gap. Additionally, we propose a completely novel set of model components tailored to address the anomaly segmentation problem. These components collectively form a new reasoning mechanism that allows iCAS to significantly outperform existing generic baselines.

Tab. 14 presents a comparison of the methods in terms of their parameters and inference time (time per image). The table highlights the key differences between the methods, specifically focusing on the number of parameters and inference speed. iCAS (Ours) stands out as the most efficient method, with only 322M parameters, roughly half the size of SAM/PerSAM and one-third the size of Matcher. Despite its smaller parameter size, iCAS achieves the fastest inference time, requiring just 0.46 seconds per image, which is approximately 1.74x faster than SAM, 3.09x faster than PerSAM, and 25.9x faster than Matcher.

| Method | Parameters (Approx.) | Inference Speed (Time per Image) |
|---|---|---|
| SAM | 641M | 0.80s |
| PerSAM | 641M | 1.42s |
| Matcher | 945M | 11.89s |
| iCAS (Ours) | 322M | 0.46s |

Table 14: Comparison of methods in terms of parameters and inference speed.

A.6 VISUALIZATION OF GQS.

Previous transformer-based methods typically rely on semantic conditions as classifiers during the query selection process. However, in anomaly detection scenarios, semantic conditions are often unavailable, posing a significant challenge for query selection. To address this, we propose a Greedy Query Selection (GQS) strategy that actively learns to select the most representative and independent query features from all image features. Notably, this selection approach is particularly well-suited for anomaly detection, as anomalies tend to exhibit distinct feature distributions. To illustrate the effectiveness of our method, we visualize several selected features in Fig 5, where GQS consistently covers all unique visual regions while ignoring redundant textures.

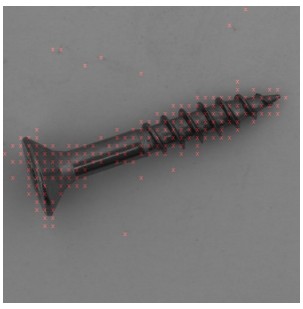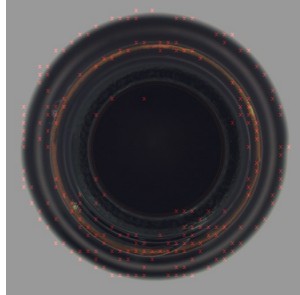

Figure 5: Visualization of selected queries by GQS mapped onto the original image.

A.7 THEORETICAL ANALYSIS OF GQS.

**Problem Formulation.** Let $\mathcal{F} = \{x_1, \ldots, x_M\}$ ($x_i \in \mathbb{R}^d$) be the set of $M$ feature vectors extracted from an input image. Our goal in the query selection phase is to select a subset of $N$ queries $\mathcal{Q} = \{q_1, \ldots, q_N\} \subset \mathcal{F}$ that best represents the feature space. We formulate this as the K-Center Problem, where the objective is to minimize the covering radius $R$:

$$R(\mathcal{Q}) = \max_{x \in \mathcal{F}} \min_{q \in \mathcal{Q}} d(x, q) \tag{2}$$

where $d(\cdot, \cdot)$ is a distance metric (e.g., Euclidean distance). Let $\mathcal{Q}^*$ denote the optimal subset that minimizes $R$, and let $R^* = R(\mathcal{Q}^*)$ be the optimal covering radius.

**Theorem 1.** *The covering radius $R_{GQS}$ produced by the Greedy Query Selection satisfies $R_{GQS} \leq 2 \cdot R^*$.*

*Proof.* Let $x_{crit}$ be the point in $\mathcal{F}$ that determines the final radius of GQS, i.e., $d(x_{crit}, \mathcal{Q}) = R_{GQS}$. Consider the set of $N + 1$ points consisting of the $N$ selected queries $\mathcal{Q} = \{q_1, \ldots, q_N\}$ plus the critical point $x_{crit}$. By the definition of the greedy strategy, for any $i < j$, the distance $d(q_j, q_i)$ must be at least the radius determined by the set $\{q_1, \ldots, q_{j-1}\}$, which is greater than or equal to the final radius $R_{GQS}$. Thus, all pairwise distances in this set of $N + 1$ points are at least $R_{GQS}$.

Now, consider the optimal clustering $\mathcal{Q}^*$ with $N$ centers. By the Pigeonhole Principle, if we distribute $N + 1$ points into $N$ clusters, at least one cluster must contain two points from our set of $N + 1$ points. Let these two points be $u$ and $v$. Since $u$ and $v$ are in the same optimal cluster with center $c^* \in \mathcal{Q}^*$, by triangle inequality:

$$d(u, v) \leq d(u, c^*) + d(v, c^*) \leq R^* + R^* = 2R^* \tag{3}$$

However, we established that the pairwise distance between any points in our set is at least $R_{GQS}$. Therefore:

$$R_{GQS} \leq d(u, v) \leq 2R^* \tag{4}$$

$\square$

Let $x_{anom} \in \mathcal{F}$ be an anomaly feature. If $x_{anom}$ is strictly distinguishable from the normal background features $\mathcal{F}_{bg}$ such that $d(x_{anom}, \mathcal{F}_{bg}) > 2R^*$, then GQS is guaranteed to select a query $q$ such that $d(x_{anom}, q) \leq 2R^*$. Unlike random sampling, which may miss sparse anomalies with a probability depending on the anomaly's size, GQS guarantees coverage as long as the anomaly is a geometric outlier in the feature space.

## A.8 VISUALIZATION

We show the results of semantic anomaly segmentation on the MVTecAD and VisA datasets for all categories in Fig. 6 and Fig. 7, with the prompt sample on the left and the segmentation results on the right. In this case, the white outline refers to the groundtruth of the anomaly region, while the blue mask indicates the predicted segmentation region. In addition, we visualize in Fig. 8 and Fig. 9 the interactive anomaly segmentation on all categories of MVTecAD and VisA, which results are generated by randomly clicking on the regions. Finally, we update the comparison with a revised version of the visualization in Fig. 10.

## A.9 FAILURE CASE

Our proposed iCAS still has failure cases for in-context anomaly segmentation. As shown in Fig. 11, for samples with multiple anomalous regions of the same type, the interactive segmentation of iCAS sometimes does not predict all regions. In addition, the current anomaly descriptions are often ambiguous, e.g., for the "color" anomaly category of MVTecAD, it includes black and red, etc. As shown in Fig. 12, the black region as an anomaly prompt has poor generalization over the red anomaly region. Therefore, making the granularity of in-context anomaly segmentation more controllable is a critical future research direction.

## A.10 LIMITATION

Different from other methods that use a few normal or abnormal samples for training, we use a certain amount of anomaly data for pre-training, as we aim to explore the pre-training of a generalized in-context anomaly segmentation model from publicly available anomaly datasets.

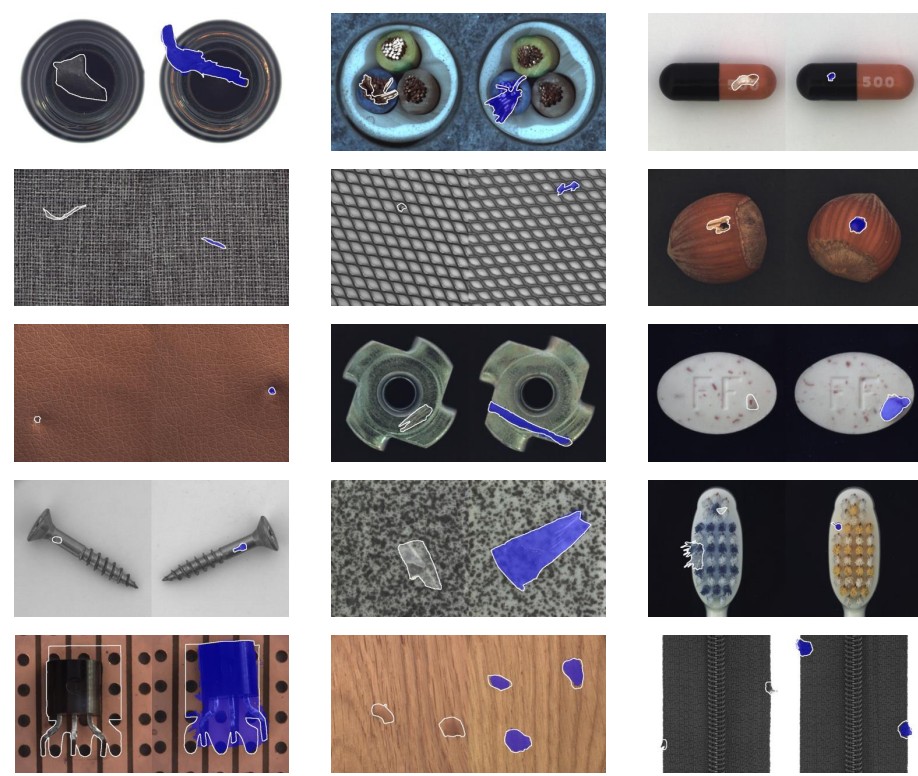

Figure 6: Visualization of semantic anomaly segmentation on MVTecAD.

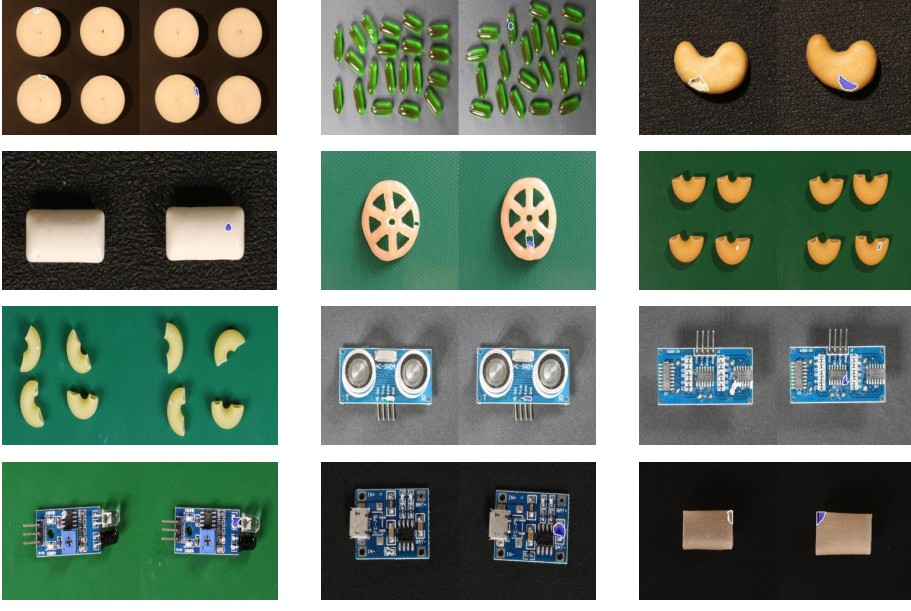

Figure 7: Visualization of semantic anomaly segmentation on VisA.

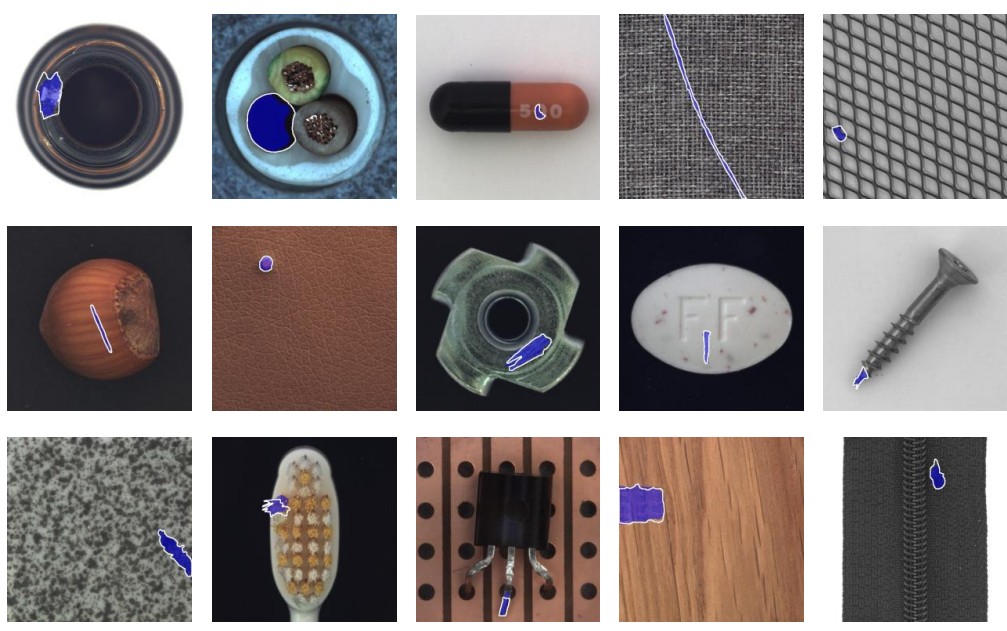

Figure 8: Visualization of interactive anomaly segmentation on MVTecAD.

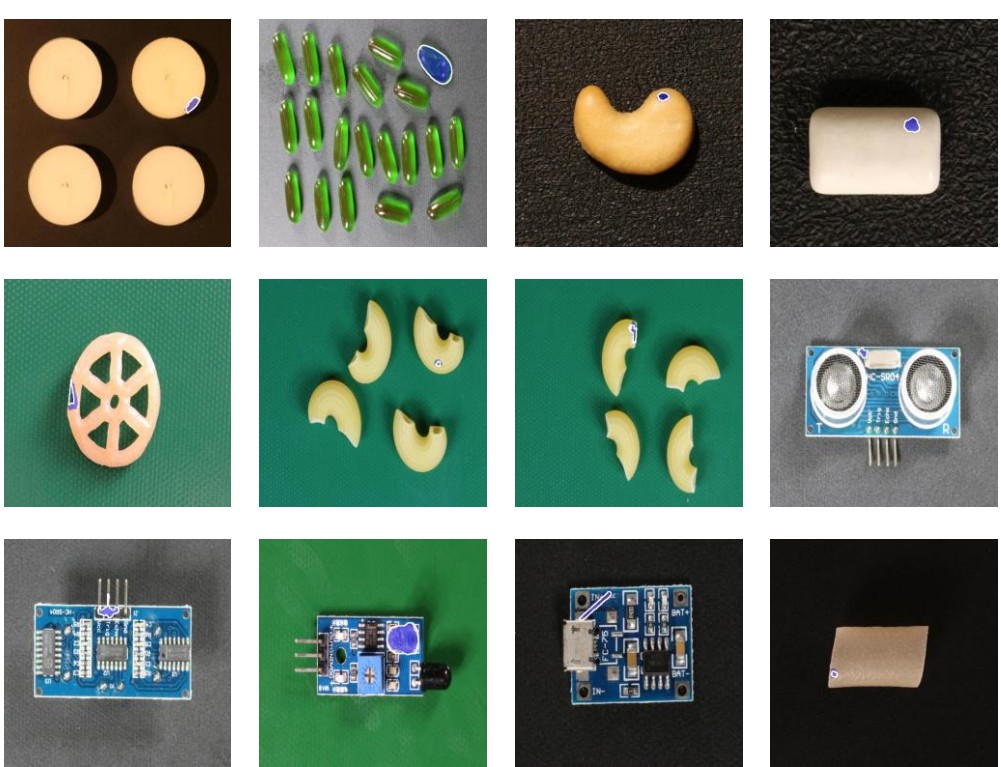

Figure 9: Visualization of interactive anomaly segmentation on VisA.

### A.11 THE USAGE OF LLMS

Given the current LLMs' outstanding performance in text tasks, we used a commercial LLM, ChatGPT, to check this paper's grammar and provide revision suggestions. No LLMs were used during research ideation or coding.

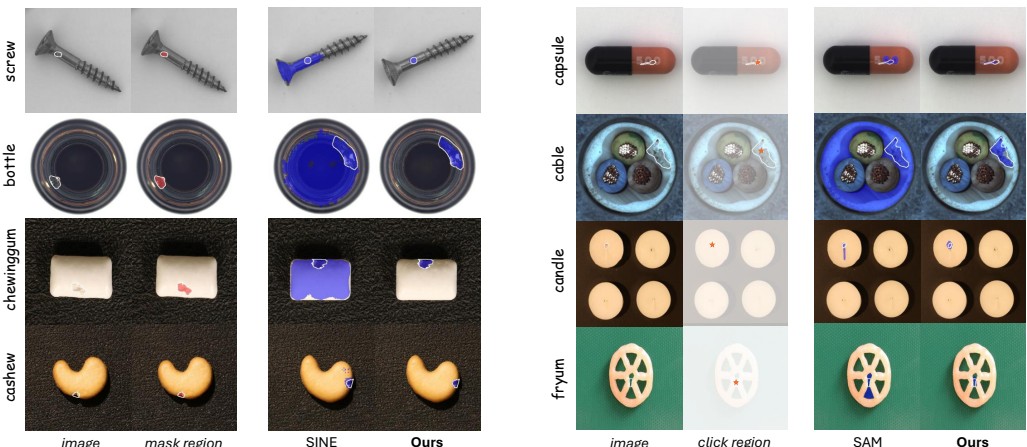

Figure 10: Qualitative comparison of in-context anomaly segmentation on MVTecAD and VisA. On the left is the semantic anomaly segmentation task compared to SINE, and on the right is the interactive anomaly segmentation task compared to SAM. We use a white contour for ground truth, a red mask for anomaly prompts, and a red star for the click region.

★ Interactive Prompting

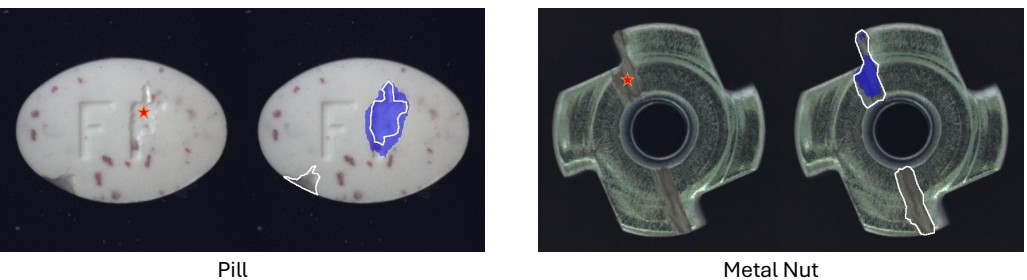

Figure 11: Failure cases of semantic anomaly segmentation

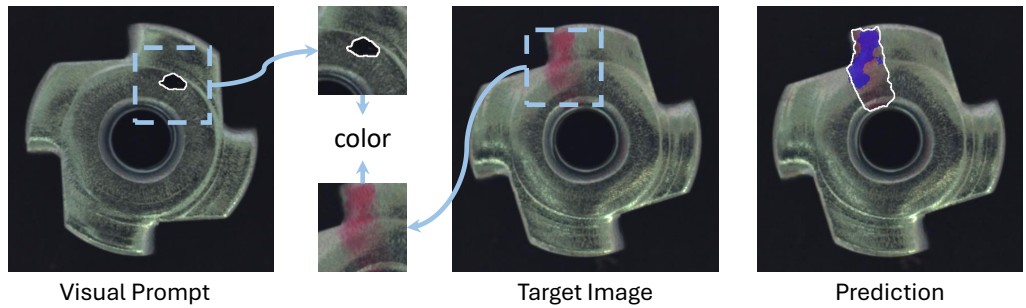

Figure 12: Failure cases of interactive anomaly segmentation