# OpenReview forum: "iCAS: A In-Context Anomaly Segmentation Framework for Industrial Visual Inspection"
_ICLR.cc/2026/Conference — Submitted to ICLR 2026_

### Official Review · Reviewer_Tqxe · 2025-10-30

**Soundness:** 3
**Presentation:** 2
**Contribution:** 4
**Rating:** 6
**Confidence:** 4

**Summary:**

This paper introduces what the authors claim to be the first in-context segmentation method tailored for industrial visual inspection. By combining a greedy query selection strategy, a mask-level feature matching module, and a general-to-specific pretraining paradigm, the proposed framework not only outperforms existing general in-context segmentation methods, but also delivers strong few-shot anomaly detection performance. Overall, the work is interesting, but there are several issues that should be addressed before it is ready for acceptance.

**Strengths:**

The paper explores the first in-context anomaly segmentation model designed specifically for industrial inspection.

The general-to-specific pretraining strategy is reasonable and clearly motivated, as training solely on industrial anomaly datasets is often insufficient.

Experimental results demonstrate promising performance.

**Weaknesses:**

The motivation needs to be strengthened. Why is an in-context segmentation model particularly important for industrial inspection? More discussion is needed.

The related work section is not comprehensive enough. Several recent methods published in 2025 are missing.

The description of the in-context transformer is too brief, making it difficult for readers to understand how it works internally.

The comparisons with PerSAM and Matcher appear unfair. The proposed method leverages both semantic segmentation data and targeted industrial anomaly data for training, while prior in-context segmentation baselines do not use anomaly detection data.

Table 3 reports promising results, but the evaluation metric used is not clearly indicated.

While I understand that space limitations make it difficult to include extensive details, the current description of the proposed method is still hard for readers to follow. Some parts require further elaboration—particularly Section 3.2 on the objective function, which is difficult to understand in its current form.

**Questions:**

How does the method perform when anomalies have fuzzy or unclear boundaries, which is very common in real industrial settings?

Figure 2 suggests that a mask set is required for training. How is this mask set obtained in practice? Does it require manual annotation?

In line 306, it is mentioned that semantic masks are obtained via SAM. How exactly is this performed?

In Table 1, what do the notations CP, BT, HN, etc. stand for? Please clarify.

---

> ### Author Response · Authors · 2025-11-26
> **(1/3) Response to Reviewer Tqxe**
>
> We sincerely thank the Reviewer Tqxe for your valuable suggestions, which help us further strengthen the motivation and clarity of our manuscript. We address each point below.
>
> ### **W1: The motivation of in-context segmentation model for industrial inspection.**
>
> Industrial inspection often involves complex and subtle anomalies, such as scratches, stains, or small defects, which can vary significantly in appearance and context. In many practical scenarios, fully supervised segmentation is impractical due to the high cost of pixel-level annotations, and purely unsupervised models (trained only on normal samples) often fail to localize anomalies precisely.
>
> An in-context segmentation model is particularly important in industrial inspection because it can adapt anomaly segmentation to different contextual sample types on the fly. Specifically, iCAS not only supports anomaly detection using a few normal samples but can also leverage a few anomaly samples, enabling few-shot adaptation. Moreover, it can provide interactive anomaly segmentation for efficient annotation, and utilize contextual semantic segmentation to deliver more precise anomaly masks. This context-aware capability allows iCAS to flexibly handle diverse anomaly types and subtle boundary variations without retraining.
>
> Large-scale pre-trained models like SAM provide strong general contextual understanding but are not specialized for anomaly boundaries. iCAS bridges this gap by combining general segmentation knowledge with anomaly-specific adaptation in an in-context framework, it enables industrial inspection systems to efficiently handle diverse anomalies, accurately delineate their boundaries, and iteratively improve detection with minimal additional data.
>
> ### **W2: The recent methods in related work.**
>
> We thank the reviewer for highlighting missing recent works from 2025. In the revised manuscript, we include these references and clarify their relation to our approach in the Related Work section.
>
> ### **W3: The description of the in-context transformer.**
>
> We appreciate your suggestion to provide more clarity on the in-context transformer. Below is a more detailed explanation of its components and mechanisms in the iCAS model.
>
> **1. Encoder: Feature Extraction**
> The iCAS model uses a pre-trained DINOv2 encoder to extract visual features from both the target image ($I_t$) and the reference image ($I_r$). The extracted feature maps ($F_t$) and ($F_r$) are passed to the next stage of the model, the in-context transformer decoder.
>
> **2. Decoder: In-context Transformer Decoder and Mask Decoder**
>
> **a. In-context Transformer Decoder**
> The in-context transformer decoder plays a crucial role in processing the feature maps ($F_t$) and ($F_r$) using various queries. These queries interact with the feature maps through multi-head self-attention and cross-attention mechanisms to generate the final anomaly segmentation masks. The decoder is responsible for performing the context-based reasoning to detect anomalies in the target image by leveraging information from the reference image.
>
> The different types of queries used in the decoder are:
> * **Content Queries ($Q_c$):** These initial queries interact with the target image feature map ($F_t$). These queries help guide the model to focus on relevant parts of the target image and segment the regions of interest for anomaly detection.
> * **Prompt Queries ($Q_p$) and ($Q_{ntp}$):** Generated from the reference image and its associated mask via mask pooling, these queries provide context about what constitutes "normal" and "anomalous" regions, helping the model refine its segmentation and distinguish between normal and abnormal regions.
> * **Foreground and Background Queries ($Q_{fg}$), ($Q_{bg}$):** These specialized queries are designed to further improve segmentation by distinguishing between the foreground (anomalous regions) and the background (normal regions), enhancing the precision of anomaly localization.
>
> **b. Mask Decoder**
> Its mask decoder is to generate a set of class-agnostic candidate masks based on the feature map of the target image ($F_t$). The mask decoder computes the dot product between the content queries ($Q_c$) and the target image feature map ($F_t$), producing $K$ potential masks ($M_t \in \mathbb{R}^{K \times H \times W}$): $M_t = \text{sigmoid}(Q_c \times F_t)$.
> These candidate masks are refined by upsampling operations to provide the final mask predictions.
>
> **c. Query Matching and Prediction**
> The predicted masks ($M_t$) and class probabilities from content queries ($Q_c$) are assigned to ground-truth segments using bipartite matching. Prompt queries ($Q_p$) and non-target prompt queries ($Q_{ntp}$) act as semantic classifiers to guide the assignment of masks to normal and anomalous regions, while foreground ($Q_{fg}$) and background ($Q_{bg}$) queries provide additional guidance to distinguish anomalous foreground from normal background.

---

> ### Author Response · Authors · 2025-11-26
> **(2/3) Response to Reviewer Tqxe**
>
> ### **W4: The comparisons with PerSAM and Matcher.**
>
> We understand the reviewer’s concern regarding PerSAM and Matcher, which were trained on the massive 11M images (SA-1B) dataset without anomaly-specific data.
>
> In contrast, iCAS uses a frozen DINOv2 backbone and trains a lightweight mask transformer on a much smaller dataset (General: 776K, Specific Anomaly: 85K). Despite a smaller training scale, iCAS outperforms the baselines due to targeted General-to-Specific training. The training details are compared below:
>
> | Method | Backbone | Params | Pre-training Data Scale | Training Strategy |
> | :--- | :--- | :--- | :--- | :--- |
> | SAM (PerSAM/Matcher) | ViT-H | 641M | SA-1B (11,000K) | Full Training |
> | iCAS (Ours) | ViT-L (DINOv2) | 322M | General 776K, Specific 85K | Frozen Backbone + Trainable In-context Transformer |
>
> This table clarifies that iCAS achieves its superior performance not through a larger model or data volume, but through a parameter-efficient, targeted training strategy that effectively injects anomaly-specific knowledge into a lightweight mask transformer.
>
> | Dataset     | Method  | mIoU  | FB-IoU |
> |:------------|:--------|:------|:-------|
> | Retinal OCT | PerSAM  | 13.6  | 41.7   |
> |             | Matcher | 8.3  | 34.2   |
> |             | iCAS    | 31.0  | 61.9   |
> | Brain MRI   | PerSAM  | 13.6  | 53.6   |
> |             | Matcher | 10.7  | 42.9   |
> |             | iCAS    | 15.1  | 55.2   |
> | Liver CT    | PerSAM  | 14.1  | 15.8   |
> |             | Matcher | 6.5  | 11.4   |
> |             | iCAS    | 47.3  | 66.8   |
>
> Our iCAS method shows fairness by consistently outperforming PerSAM and Matcher across multiple medical datasets, including Retinal OCT, Brain MRI, and Liver CT. Importantly, during training, iCAS was not exposed to any domain-specific data from these medical fields, ensuring that the model’s strong performance is not biased by prior knowledge of the datasets. This demonstrates iCAS's robust and equitable ability to generalize across diverse medical imaging tasks, providing a fair and unbiased evaluation of model performance.
>
> ### **W5: The evaluation metric in Table 3.**
>
> The evaluation metric in Table 3 is AUROC, a standard in anomaly detection. We will clarify this in the table caption.
>
> ### **W6: The detailed description of the proposed method's objective function.**
>
> The objective function of the iCAS model consists of three main components aimed at minimizing the discrepancy between the predicted segmentation masks and the ground-truth masks for both normal and anomalous regions.
>
> Given a set of $K$ predicted probability-mask pairs ($z = \{(p_i, M_i)\}\_{i=1}^{K}$) and $K_t$ ground-truth segments ($z_t = \{(c_i, M_i^t)\}\_{i=1}^{K_t}$), the objective function is defined as:
>
> $$
> L_{\text{ic}}(z, z_t) = \sum_{i=1}^{K} [-\log p_i(c_j^t) - \log p_{ntp}(c_j^t) + L_{\text{mask}}(M_i, M_i^t)]
> $$
>
> The loss function consists of three parts:
>
> * The first term, $-\log p_i(c_j^t)$, is a cross-entropy term corresponding to the likelihood that the predicted mask belongs to the correct class (normal or anomalous).
> * The second term, $-\log p_{ntp}(c_j^t)$, is a cross-entropy term corresponding to the likelihood of the non-target prompt mask. This term helps the model recognize content that deviates from the non-target context, which is crucial for identifying anomalies. These terms together ensure the model effectively distinguishes between normal and abnormal regions based on the context provided.
> * The third term, $L_{\text{mask}}(M_i, M_i^t)$, is the binary mask loss function that measures the pixel-wise difference between the predicted mask ($M_i$) and the ground-truth mask ($M_i^t$). Typically, this is calculated using binary cross-entropy loss:
>     $$
>     L_{\text{mask}}(M_i, M_i^t) = - (M_i^t \cdot \log(M_i) + (1 - M_i^t) \cdot \log(1 - M_i))
>     $$
> This loss term helps align the predicted mask with the true anomaly or normal regions in the image.

---

> ### Author Response · Authors · 2025-11-26
> **(3/3) Response to Reviewer Tqxe**
>
> ### **Q1: How does the method perform when anomalies have fuzzy or unclear boundaries?**
>
> General segmentation models like SAM and SINE often struggle with subtle or fuzzy anomalies because their full-image mask sets tend to merge small defects with the overall object, failing to capture precise boundaries. In contrast, iCAS benefits from training on anomaly-specific datasets, which enhances sensitivity to irregular boundaries. This allows iCAS to better distinguish anomalous regions from normal areas, producing tighter and more accurate masks even when anomalies have unclear edges. The qualitative results in Figure 4 clearly demonstrate this improvement.
>
> ### **Q2: Figure 2 suggests that a mask set is required for training. How is this mask set obtained in practice? Does it require manual annotation?**
>
> We would like to clarify a misunderstanding. The "mask set" in Figure 2 is generated by the mask decoder and is class-agnostic, not externally provided. During the general training phase, the mask decoder learns to predict masks for all regions of an image, inspired by MaskFormer and MM-Former. For each image, it outputs a set of candidate masks, which are then assigned to target regions (e.g., anomalies) via conditional matching. We further enhance this general mask set using anomaly-specific data to improve the model’s ability to extract masks for anomalous regions. In the Multi-Feature Matching (MFM) stage, these mask sets are used to compare normal and abnormal region-level features and refine anomaly detection.
>
> ### **Q3: In line 306, it is mentioned that semantic masks are obtained via SAM. How exactly is this performed?**
>
> To enrich our pre-training dataset, we leverage SAM’s general segmentation capabilities to generate masks for the Objects365 dataset, which only provides bounding boxes. Specifically, SAM can produce accurate masks given bounding boxes, allowing us to create high-quality pseudo-mask annotations without manual pixel-level labeling. This is a practical dataset generation strategy while maintaining mask quality for generalization.
>
> ### **Q4: In Table 1, what do the notations CP, BT, HN, etc. stand for?**
>
> The abbreviations CP, BT, HN, etc., correspond to object categories in the MVTecAD dataset (e.g., Carpet, Bottle, Hazelnut). Due to space limitations, we used short labels in the main table. The full nomenclature is provided in the **Appendix A.2**, and in the revised manuscript we will explicitly clarify these abbreviations in the table caption for completeness.
> We apologize for the clarity in the manuscript and we will highlight this indication in the manuscript.
>
> ### Summary:
> **We are deeply grateful to Reviewer Tqxe for acknowledging our innovation, training strategy, and experiments. Based on Reviewer Tqxe's constructive feedback, we have made the following revisions in blue font within the manuscript:**
> 1. **We emphasized our motivation in Lines 36-43.**
> 2. **We supplemented recent studies in Lines 136-140.**
> 3. **We added metric descriptions to Tables 3 and 4.**
> 4. **We have supplemented fairness experiments in the medical imaging domain in Tables 5 and 6, with explanations provided in lines 464-473.**
> 5. **Full category names are detailed in lines 348-349 and Appendix A.2.**

---

### Official Review · Reviewer_eNU5 · 2025-10-31

**Soundness:** 3
**Presentation:** 3
**Contribution:** 2
**Rating:** 4
**Confidence:** 3

**Summary:**

This paper proposes the iCAS model together with a General-to-Specific pre-training paradigm. iCAS is based on a mask classification transformer architecture and introduces Greedy Query Selection (GQS) and Mask-level Feature Matching (MFM) to accurately localize anomalous regions. This approach enables robust anomaly segmentation, even under limited anomaly data conditions.

**Strengths:**

1. This paper evaluates the iCAS model using a wide range of evaluation metrics and a variety of well-structured experiments.
2. This paper defines a new pre-training paradigm, the General-to-Specific approach, that effectively bridges the gap between general semantic segmentation and specialized anomaly segmentation.

**Weaknesses:**

1. The paper lacks comparison with recent few-shot anomaly detection methods, such as: UniVAD: A Training-free Unified Model for Few-shot Visual Anomaly Detection (CVPR 2025), DictAS: A Framework for Class-Generalizable Few-Shot Anomaly Segmentation via Dictionary Lookup (ICCV 2025)

2. Experiments on backbone networks are limited. Ablation studies involving CLIP-ViT and DINOv1 would strengthen the effectiveness of this paper.

3. Greedy Query Sampling (GQS) is likely sensitive to the choice of K, but no ablation studies on this parameter have been presented. Furthermore, if GQS tends to select queries in the normal region, the robustness of the model should be verified on datasets with diverse object locations, such as MPDD or medical datasets, which are not covered in this paper.

4. General-to-Specific pre-training paradigm consists of two stages; an analysis of its computational cost would be valuable.

**Questions:**

1. We are curious about the performance of iCAS in an unsupervised learning anomaly detection environment and its performance in the presence of anomaly data.
2. For backbone networks, a comparison study comparing DINOv2, CLIP-ViT, and DINOv1 would be interesting.
3. General-to-Specific pre-training paradigm consists of two stages, and an analysis of their computational costs would be important.

---

> ### Author Response · Authors · 2025-11-26
> **(1/2) Response to Reviewer eNU5**
>
> We would like to greatly thank the Reviewer eNU5 for your comments.  Following the comments, we have enhanced the completeness of the manuscript and improved the reliability of the methodology by adding additional experiments. The details are as follows:
>
> ### **W1: Lack of comparison with recent few-shot anomaly detection methods.**
>
> We have included comprehensive comparisons with DictAS and UniVAD in the revised tables. Both DictAS and UniVAD are strong recent baselines, but their design philosophies differ substantially from that of iCAS, which helps explain the observed behavior across benchmarks.
>
> | Method | MVTecAD Sample AUROC | MVTecAD Pixel AUROC | MVTecAD mIoU | VisA Sample AUROC | VisA Pixel AUROC | VisA mIoU |
> | :--- | :--- | :--- | :--- | :--- | :--- | :--- |
> | DictAS | 97.5 | 98.6 | 25.6 | 91.7 | 98.8 | 7.8 |
> | iCAS | 97.8 | 97.3 | 51.1 | 93.0 | 98.1 | 37.3 |
>
> **DictAS:** DictAS is specifically optimized to maximize segmentation AUROC in unsupervised or few-shot settings. Accordingly, it achieves high pixel-level AUROC. However, its threshold-sensitive formulation results in notably lower mIoU on both MVTecAD and VisA. In contrast, iCAS maintains competitive AUROC while substantially improving mIoU. This reflects a key methodological difference: DictAS focuses on a single task formulation, whereas iCAS is designed as a broadly applicable anomaly segmentation framework that naturally supports interactive anomaly segmentation, semantic anomaly segmentation, few-shot AD, and open-set AD within a unified model architecture. As such, iCAS aims for balanced, stable performance across multiple AD scenarios rather than optimizing for one metric.
>
> | Method | MVTecAD Sample AUROC | MVTecAD Pixel AUROC | VisA Sample AUROC | VisA Pixel AUROC | BrainMRI Sample AUROC | BrainMRI Pixel AUROC | LiverCT Sample AUROC | LiverCT Pixel AUROC | RESC Sample AUROC | RESC Pixel AUROC |
> | :--- | :--- | :--- | :--- | :--- | :--- | :--- | :--- | :--- | :--- | :--- |
> | UniVAD | 97.8 | 96.5 | 93.5 | 98.2 | 80.2 | 96.8 | 70.0 | 96.3 | 85.5 | 94.9 |
> | iCAS | 97.5 | 96.9 | 92.6 | 98.0 | 91.3 | 96.6 | 74.9 | 98.1 | 84.2 | 92.4 |
>
> **UniVAD:** UniVAD follows a model-ensemble paradigm, integrating several large pretrained models (GroundingDINO, RAM, SAM, CLIP, DINOv2) to leverage complementary cross-model cues. In contrast, iCAS is built as a single unified context model trained from scratch, without external detectors, segmentors, or vision-language models. The two systems therefore differ not only in complexity but also in intended use cases: UniVAD explores the benefits of model cooperation, while iCAS emphasizes generality, scalability, and deployment simplicity.
>
> ### **W2: Experiments on backbone networks are limited.**
>
> Thank you for your constructive suggestion to expand backbone ablation studies, this helps further validate our backbone selection rationale and strengthens the paper’s experimental rigor. We have conducted additional experiments on CLIP-ViT-Large and DINOv1-Large under the exact same setup as our original semantic anomaly segmentation evaluation.
>
> | Backbone Model | MVTecAD mIoU (%) | MVTecAD FB-IoU (%) | VisA mIoU (%) | VisA FB-IoU (%) |
> | :--- | :--- | :--- | :--- | :--- |
> | CLIP-ViT-Large | 40.9 | 68.2 | 31.2 | 62.2 |
> | DINOv1-Large | 45.4 | 72.9 | 33.9 | 65.4 |
> | DINOv2-Large | 51.1 | 74.9 | 37.3 | 68.5 |
>
> DINOv2 consistently outperforms DINOv1 under the same setup, thanks to larger-scale pre-training and more effective contrastive/self-supervised learning, which significantly enhances local semantic representation, making it more capable of detecting subtle, small-scale anomalies. In contrast, CLIP-ViT emphasizes global image-text alignment, resulting in weaker local semantic understanding and lower segmentation performance.

---

> ### Author Response · Authors · 2025-11-26
> **(2/2) Response to Reviewer eNU5**
>
> ### **W3: GQS sensitivity to K and robustness on diverse datasets.**
>
> To address the concern regarding the sensitivity of GQS to the number of queries (K), we performed an ablation study on the MVTecAD dataset.
>
> | Setting | K Queries | mIoU (%) | FB-IoU (%) |
> | :--- | :--- | :--- | :--- |
> | Interactive | 50 | 59.2 | 81.2 |
> | | 100 | 59.7 | 81.7 |
> | | 150 | 61.3 | 82.2 |
> | | 200 (Default) | 63.7 | 83.8 |
> | Semantic | 50 | 47.2 | 73.8 |
> | | 100 | 50.3 | 74.1 |
> | | 150 | 50.9 | 74.3 |
> | | 200 (Default) | 51.1 | 74.9 |
>
> As shown in the table, the performance of iCAS remains highly stable. Even when reducing K to 50, the Interactive mIoU only drops marginally, indicating that GQS efficiently captures representative features early in the sampling process and is not overly sensitive to the specific choice of K.
>
> Regarding the concern that GQS might favor normal regions, we clarify that our method performs representative sampling in the feature space rather than the spatial domain. By formulating query selection as a $k$-center problem, GQS minimizes the maximum distance between any image feature and the selected queries. Since anomalies typically manifest as distinct outliers in the feature space, the algorithm is inherently driven to select these outlier features to satisfy the covering bound, thereby ensuring that both normal and anomalous patterns are effectively captured.
>
> To verify the robustness of our model on datasets with diverse object locations and structures (beyond industrial inspection), we evaluated iCAS on three medical segmentation datasets:
> | Dataset | Method | mIoU | FB-IoU |
> | :--- | :--- | :--- | :--- |
> | Retinal OCT | SAM | 43.5 | 63.7 |
> |  | iCAS (Ours) | 54.5 | 72.0 |
> | Brain MRI | SAM | 21.0 | 52.7 |
> |  | iCAS (Ours) | 49.4 | 72.9 |
> | Liver CT | SAM | 20.5 | 58.4 |
> |  | iCAS (Ours) | 42.3 | 70.6 |
>
> These results confirm that iCAS is not limited to specific industrial setups but is robust across diverse domains with varying object locations and structures.
>
> ### **W4: Computational cost analysis of General-to-Specific pre-training paradigm.**
>
> We appreciate the reviewer's suggestion to analyze the computational cost. While our General-to-Specific paradigm involves a comprehensive training phase (totaling 3,625K iterations), we contend that the computational cost is justifiable and parameter-efficient compared to foundation model pre-training.
>
> | Method | Backbone Architecture | Params (Total / Trainable) | Pre-training Data Scale | Training Strategy | Iteration |
> | :--- | :--- | :--- | :--- | :--- | :--- |
> | SAM | ViT-H | 641M | SA-1B (11,000K) | Full Training | 11,000K |
> | iCAS | ViT-L (DINOv2) | 322M | Genral 776K, Specific 85K (861K) | Frozen Backbone + Trainable MaskTransformer | 3,625K |
>
> Unlike SAM, which performs computationally intensive full-parameter updates on a massive ViT-H backbone (641M params) across 11,000K iterations, iCAS adopts a Frozen Backbone strategy (ViT-L), restricting gradient computation solely to the in-context mask transformer. This design drastically reduces the memory footprint and FLOPs per step. Furthermore, our total training iterations are approximately 1/3 of SAM's exposure, utilizing only 7.8% of the unique data scale, demonstrating high data and cost efficiency relative to the performance achieved.
>
> ### Summary:
> **We are particularly grateful to Reviewer eNU5 for their insightful suggestions regarding our experiments and comparisons. Based on the discussion, we have conscientiously supplemented these experiments in the revised manuscript in blue font:**
> 1. **We introduced DictAS and UniVAD as recent relevant work in Lines 136-140.**
> 2. **We have added comparative experiments with CLIP and Dinov1 in Appendix Table 11 and Lines 858–863.**
> 3. **We present query selection ablation studies and analysis in Table 12 and Lines 898–902.**
> 4. **We provide a detailed analysis of iCAS training and inference costs in Appendix A.5.**

---

### Official Review · Reviewer_QgxX · 2025-10-31

**Soundness:** 3
**Presentation:** 3
**Contribution:** 3
**Rating:** 6
**Confidence:** 4

**Summary:**

This paper introduces iCAS, a new In-Context Anomaly Segmentation framework designed to generalize visual in-context prompting (e.g., SAM-style models) to industrial anomaly segmentation. The core idea is to enable training-based, prompt-driven anomaly localization using only a few anomaly or normal samples. The method achieves strong performance across diverse datasets, significantly outperforming existing in-context segmentation models and anomaly detection methods.

**Strengths:**

1.iCAS unifies promptable segmentation (SAM-like) and anomaly detection in a training-based, in-context fashion

2.The proposed GQS and MFM modules are simple yet effective, addressing practical issues of query redundancy and boundary precision.

3.Extensive experiments (five datasets, multiple tasks, and ablations) convincingly show robustness, scalability, and effectiveness of each component.

**Weaknesses:**

1.While iCAS performs well, its components (GQS, MFM, two-stage training) are mostly adapted from known concepts (active learning, mask matching, transfer learning). The true methodological novelty might be seen as moderate.

2.The paper primarily focuses on industrial surface defects, it is unclear whether iCAS generalizes to other anomaly types (e.g., medical or natural images) or remains domain-specific due to the specialized anomaly-aware pre-training.

**Questions:**

1.Could the authors clarify why GQS is necessary beyond standard query embeddings? How much does it reduce redundancy compared to random or uniform query sampling?

2.How scalable is the method computationally? Please report training cost and inference time compared to SAM or Matcher.

---

> ### Author Response · Authors · 2025-11-26
> **(1/2) Response to Reviewer QgxX**
>
> We appreciate the reviewer's thorough acknowledgment of the motivation behind the proposed iCAS and the validity of our methodology. In response to the thoughtful and detailed feedback provided, particularly the critical inquiries regarding the novelty, generalizability, and computational efficiency of our research approach, we address each point as follows:
>
> ### **W1: Components are mostly adapted from known concepts.**
>
> While components like active learning or mask matching exist in other fields, their adaptation in iCAS addresses fundamental failure modes of previous studies when applied to industrial inspection.
>
> Regarding the necessity of Greedy Query Selection (GQS), we explicitly position it as a specialized adaptation for transformer-based detectors to enhance the selection of potential anomalous queries. This capability is unattainable by standard Random or Class-Conditional strategies. Random sampling faces an efficiency bottleneck for sparse defects, as the number of queries required to achieve high confidence becomes prohibitively large for fine-grained anomalies. Class-conditional selection relies on high classification scores, an assumption that fails in UAD since anomalies are undefined unknowns, leading to their suppression. To bridge this gap, we formulate GQS geometrically as the k-center problem. By minimizing the covering radius, GQS provides a theoretical 2-approximation guarantee ($R_{GQS} \le 2 R^*$), ensuring that any feature point, including outliers, is mathematically constrained to be covered as we indicate in Appendix A.6&A.7.
>
> The MFM module is theoretically anchored in two core principles aligned with anomaly segmentation’s demands. First, to capture fine-grained structural deviations of industrial anomalies, MFM uses mask pooling to aggregate features within mask regions, converting pixel-level features into mask-level embeddings. This design directly addresses a key limitation of generic pixel or token-level matching, which often fails to capture subtle structural anomalies. Second, MFM’s scoring logic ties directly to our in-context learning objective, allowing it to adapt to diverse inference scenarios.
>
> Finally, our General-to-Specific strategy is not conventional transfer learning. It is a strategic design to train a generalist In-Context Anomaly Segmentation model capable of performing multiple tasks within a single framework. The two stages establish robust open-world representations and then performing specialized anomaly-aware adaptation, which are critical for enabling training-free generalization, a capability that single-stage training strategies cannot achieve.
>
> ### **W2: Generalization beyond industrial defects is unclear.**
>
> To directly evaluate whether iCAS generalizes beyond industrial defects, we additionally tested the model on three medical anomaly segmentation datasets covering very different imaging modalities: Retinal OCT, Brain MRI, and Liver CT. We compare iCAS against SAM, which is a strong domain-agnostic baseline.
>
> | Dataset     | Method | mIoU  | Δ mIoU | FB-IoU | Δ FB-IoU |
> |-------------|--------|-------|--------|--------|----------|
> | Retinal OCT | SAM    | 43.5  | –      | 63.7   | –        |
> |             | iCAS   | 54.5  | +11.0  | 72.0   | +8.3     |
> | Brain MRI   | SAM    | 21.0  | –      | 52.7   | –        |
> |             | iCAS   | 49.4  | +28.4  | 72.9   | +20.2    |
> | Liver CT    | SAM    | 20.5  | –      | 58.4   | –        |
> |             | iCAS   | 42.3  | +21.8  | 70.6   | +12.2    |
>
>
> Across all three datasets, iCAS surpass SAM by 11～28 mIoU and 8～20 FB-IoU, despite the large domain shift and the fact that medical data were never used during anomaly-aware pretraining. These results suggest that the General-to-Specific training and the design of iCAS do not overfit to industrial surfaces. Instead, they successfully improve the model’s anomaly discrimination ability in a domain-agnostic manner.

---

> ### Author Response · Authors · 2025-11-26
> **(2/2) Response to Reviewer QgxX**
>
> ### **Q1: Clarification on GQS necessity and redundancy reduction.**
>
> We propose Greedy Query Selection (GQS) to resolve a critical conflict between the evolution of Transformer detectors and the unique constraints of anomaly detection. We position GQS by analyzing the transition from standard DETR to modern two-stage detectors (e.g., Deformable DETR, DINO, MaskDINO) regarding convergence efficiency.
>
> Learned queries (standard DETR) suffer from slow convergence because they are static and content-agnostic. Class-conditional query selection (two-stage model) accelerates convergence by selecting high-scoring proposals, but this is inapplicable to anomaly detection because anomalies are undefined "unknowns" with low objectness scores, leading to their being filtered out as background. Random Sampling, while unbiased, fails statistically on sparse anomalies. For a minute defect, sampling $N=200$ queries yields a high probability ($\sim 86\\%$) of missing the defect, resulting in high redundancy.
>
> GQS bridges this gap by adopting the two-stage philosophy of initializing from image content to accelerate convergence, but replaces the supervised classification score with an unsupervised geometric distance. By formulating this as a k-center problem, GQS ensures that distinct outliers are selected to satisfy the covering bound ($R \le 2R^*$), offering the adaptability of modern detectors without needing semantic labels, as we discussed in Appendix A.6&A.7.
>
> The following table provides a theoretical comparison of these mechanisms:
>
> | Selection Strategy | Selection Mechanism | Impact on Convergence & Anomaly Detection |
> | :--- | :--- | :--- |
> | Learned Queries (e.g., DETR) | $Q = \theta_{learnable}$ | Slow Convergence due to Global Priors. |
> | Class-Conditional (e.g., Deformable DETR) | $S = \text{TopK}_{x} \big( P(\text{class} \| x) \big)$ | Inapplicable to AD (Supervision Mismatch). |
> | Random Sampling (Unbiased Baseline) | $P(\text{Miss}) \approx (1 - \frac{1}{M})^N$ | Statistical Inefficiency, leads to high redundancy. |
> | GQS (Ours) | $R_{GQS} \le 2\cdot R^*$ | Better Convergence via Deterministic Coverage. |
>
> ### **Q2: Computational scalability and inference cost comparison.**
>
> The following table compares iCAS with other models in terms of computational scalability and functional versatility:
>
>
> | Method      | Backbone Architecture        | Parameters | Pre-training Data Scale | Inference Speed (Time per Image) | Interactive Anomaly Seg. | Semantic Anomaly Seg. | Few-shot AD | Open-set AD |
> | ----------- | ---------------------------- | -------------------- | ----------------------- | -------------------------------- | ------------------------ | --------------------- | ----------- | ----------- |
> | SAM         | ViT-H                        | 641M                 | SA-1B (11,000K)         | 0.80s                            | ✓                        | ✗                     | ✗           | ✗           |
> | PerSAM      | ViT-H (SAM)                  | 641M                 | SA-1B (11,000K)         | 1.42s                            | ✗                        | ✓                     | ✗           | ✗           |
> | Matcher     | ViT-H (SAM) + ViT-L (DINOv2) | 945M                 | SA-1B (11,000K)         | 11.89s                            | ✗                        | ✓                     | ✗           | ✗           |
> | iCAS  | ViT-L (DINOv2)               | 322M                 | 861K                    | 0.46s                            | ✓                        | ✓                     | ✓           | ✓           |
>
> iCAS is significantly more lightweight and efficient than the compared SAM, PerSAM and Matcher models. It uses only 322M parameters, roughly half the size of SAM/PerSAM and one-third the size of Matcher. Furthermore, iCAS achieves 0.46s per image for inference, making it approximately 1.74x faster than SAM (0.80s per image), 3.09x faster than PerSAM (1.42s per image), and 25.9x faster than Matcher (11.89s per image). This high speed is critical for real-time industrial inspection. Finally, the total training samples (861K) are substantially smaller than those used for SAM and Matcher (11,000K), demonstrating superior data efficiency.
>
> ### Summary:
> **We are greatly appreciative of the valuable suggestions and feedback provided by Reviewer QgxX. Based on the reviewer's comments, we have made the following additions to the manuscript in blue font:**
> 1. **We have supplemented Appendix A.6 and A.7 with further analysis of GQS within our methodology.**
> 2. **We have added comparative experiments on additional medical datasets, as shown in Tables 5 and 6, along with the analysis in lines 464-473.**
> 3. **We have included an analysis and comparison of training and inference costs in Appendix A.5.**

---

### Official Review · Reviewer_AgJu · 2025-11-01

**Soundness:** 3
**Presentation:** 3
**Contribution:** 2
**Rating:** 4
**Confidence:** 3

**Summary:**

The paper proposes iCAS, an in-context anomaly segmentation framework designed for industrial visual inspection.  The model extends a mask-classification transformer with two modules: Greedy Query Selection (GQS), which selects representative visual tokens, and Mask-level Feature Matching (MFM), which refines mask alignment across queries.  It further adopts a General-to-Specific pre-training schedule—first training on large-scale semantic segmentation datasets, then fine-tuning on anomaly-focused data (RealIAD, MANTA).  Experiments on MVTec-AD, VisA, and related benchmarks report improved mIoU compared to existing in-context and SAM-based baselines (PerSAM, Matcher, SINE).

**Strengths:**

•	Ambitious effort to unify semantic, interactive, and few-shot anomaly segmentation within one framework.
•	Broad empirical evaluation across multiple benchmarks with consistent results.
•	Well-executed ablation studies showing measurable effects of GQS, MFM, and the pre-training strategy.
•	Built on open, reproducible components (MaskFormer, DINOv2), which aids transparency.

**Weaknesses:**

1. The proposed method is primarily a reassembly of existing techniques with limited conceptual advancement.
2. Claims of “training-free” or “in-context reasoning” are overstated, given the reliance on large-scale supervised pre-training and conventional fine-tuning.
3. The contributions of GQS and MFM are empirically validated but not theoretically grounded or well-explained.
4. The paper does not clearly separate iCAS from prior in-context segmentation approaches (SAM, HQ-SAM, SegGPT, SINE, Matcher). GQS and MFM are minor technical variations on existing promptable frameworks and do not introduce a new capability or reasoning mechanism.
5. Existing vision-language anomaly detection models (WinCLIP, AnomalyCLIP, InCTRL, RegAD, MetaUAS) are mentioned but never compared under equal conditions. The reported gains may stem from backbone choice and training scale rather than a new formulation.
6. The General-to-Specific pre-training scheme closely follows the standard “pretrain on generic segmentation → fine-tune on anomaly masks” pipeline already used in RegAD, MetaUAS, and RealNet. Thus, its novelty is marginal.

**Questions:**

This paper presents a solid empirical study with clear organization and credible results, but the core contribution lies in engineering integration rather than conceptual innovation.

---

> ### Author Response · Authors · 2025-11-26
> **(1/4) Response to Reviewer AgJu**
>
> We appreciate your feedback, particularly your recognition of our concept for unifying multiple tasks through contextual anomaly segmentation. However, we respectfully believe there may be a misunderstanding regarding our methodology, which led to concerns about its novelty. We provide a detailed clarification below.
>
> ### **W1: Technical Differences and Conceptual Advancement.**
> While we acknowledge building upon established backbones (e.g., DINOv2, Transformer decoders), **our core contribution lies in identifying why these generic components fail in anomaly segmentation tasks and redesigning them specifically for industrial inspection**.
> Indeed, similar to how foundational in-context models like SAM and SegGPT built upon stable transformer architectures while focusing innovation on their training paradigms and in-context learning strategies, iCAS follows this established and effective approach. The core novelty of our work lies not in reinventing the base architecture but in designing a specialized in-context learning framework that enables a generic transformer to excel at the specific task of anomaly segmentation.
> This framework comprises two key elements:
>
> **A Novel Training Paradigm**: We designed a unique set of learnable queries (foreground/background, prompt queries) and the General-to-Specific pre-training strategy. This is the cornerstone that teaches the model to perform in-context reasoning for anomaly segmentation, analogous to how SAM's "promptable" training enables it to segment anything.
>
> **Targeted Enhancements for Anomalies**: Within this paradigm, we introduced GQS and MFM as specialized components to address the specific failure modes of generic models on fine-grained defects. GQS ensures sparse anomalies are not overlooked during inference, and MFM provides the granularity needed for precise region-level comparison.
>
> Overall, the conceptual contribution is the anomaly segmentation-specific in-context learning framework. GQS and MFM are not mere incremental tweaks but are essential, targeted innovations within this framework, enabling robust, training-free anomaly segmentation that existing models cannot achieve.
>
> ### **W2: The Claims of “training-free” or “in-context reasoning”.**
>
> We understand the reviewer’s concern regarding the claims of “training-free” and “in-context reasoning”. This issue primarily stems from the need to distinguish between the necessary pre-training foundation and the resulting operational capability of the model. We declare that General-to-Specific is pre-trained on the source dataset, while in-context and training-free inference are performed on
> novel target datasets.
> We clarify our usage, which aligns with the established lexicon of foundation models (e.g., SAM):
>
> **"Training-Free"**: The term "training-free" refers strictly to the inference phase of our model.
> Like SAM and SegGPT, iCAS undergoes a single, extensive pre-training phase. However, after this initial pre-training, the model is fixed and requires no further fine-tuning or weight updates to perform anomaly segmentation on novel industrial datasets or unseen defect types. This contrasts with existing anomaly detection methods, which often require per-dataset fitting or training. iCAS's ability to generalize immediately after pre-training is what we position as "training-free" inference.
>
> **"In-Context Reasoning"**: We define this as the model's ability to dynamically adapt its segmentation logic based on the provided visual prompts (e.g., a normal image, an anomaly mask, or a click) without altering its parameters. The reliance on large-scale pre-training is precisely what enables this emergent capability, a principle well-established by foundational models like SAM and SegGPT.
>
> The pre-training is the necessary foundation, while "training-free" and "in-context" describe the model's unique and efficient operational mode post-training.
> We will revise the paper to ensure the definitions of "training-free" and "in-context reasoning" are explicitly clarified to prevent this ambiguity.

---

> ### Author Response · Authors · 2025-11-26
> **(2/4) Response to Reviewer AgJu**
>
> ### **W3: The heoretical grounding/explanation of GQS/MFM.**
>
> **GQS vs. Existing Query Selection Methods**:
> We explicitly position GQS as a specialized adaptation designed for transformer-based detectors to enhance the selection of potential anomalous queries directly from image features. This capability actively targeting deviation without supervision is unattainable by standard Random or Class-Conditional strategies due to the unique constraints of the anomaly detection task.
>
> Random selection (as used in standard DETR) is a valid unbiased strategy. However, in the specific context of industrial inspection, it faces an efficiency bottleneck due to the extreme sparsity of defects. Let $V_{total}$ be the total feature volume and $V_{anom}$ be the volume of the anomaly. Defining sparsity as $\epsilon = V_{anom} / V_{total}$, the probability of a random query set covering the anomaly is $P(\text{coverage}) \approx 1 - e^{-N\epsilon}$. While this approach eventually converges, for fine-grained defects where $\epsilon \to 0$, the number of queries $N$ required to achieve a high confidence level becomes prohibitively large. Therefore, compared to active selection strategies, random selection or initialization tends to be less effective at capturing subtle anomalies.
>
> Class-conditional selection (as seen in DINO/MaskDINO) relies on the premise that targets of interest exhibit high classification confidence. This assumption holds well for semantic segmentation but does not apply to the anomaly detection setting. In UAD, defects (e.g., scratches, stains) are not predefined semantic categories and often possess low objectness scores compared to the dominant background object. Consequently, the supervision signal required to drive class-conditional selection is absent. Without this prior, such mechanisms lack the necessary guidance to prioritize anomalous regions over salient normal features.
>
> We propose GQS (Greedy Query Selection) to bridge this gap. Since we cannot rely on class priors (precluding class-conditional) and seek higher efficiency than random sampling or initializing, we formulate the problem geometrically as the Metric K-Center Problem.
>
> By minimizing the covering radius, GQS provides a theoretical 2-approximation guarantee($R_{GQS} \le 2 R^\*$). This ensures that any feature point is deterministically covered if it lies within the feature manifold. Specifically, if an anomaly is an outlier (distance$> 2R^\*$), GQS is mathematically constrained to select a query near it to satisfy the bound. This allows iCAS to select sparse, unknown anomalous queries with high data efficiency and without requiring the semantic supervision that is unavailable in the anomaly detection task.
>
> **Analysis of MFM**:
> The Mask-MFM module is based on two core principles aligned with anomaly segmentation needs, both of which are discussed in our manuscript.
>
> First, industrial anomalies like subtle cracks and low-contrast deformations are characterized by fine-grained structural deviations. To capture these, MFM uses mask pooling, which aggregates features within candidate mask regions to convert pixel-level features into mask-level embeddings that capture structural semantics. This overcomes the limitation of pixel or token-level matching, which often fails to detect such anomalies in prior models.
>
> Second, MFM’s scoring logic aligns with our in-context learning objective. For anomaly prompting, it computes similarity between the target image’s mask embeddings and the anomaly prompt’s embeddings, consistent with how we define prompt queries for semantic inference. For normal prompting, it uses dissimilarity between target mask embeddings and normal reference embeddings, drawing on the negative likelihood principle behind our normal prompting anomaly score

---

> ### Author Response · Authors · 2025-11-26
> **(3/4) Response to Reviewer AgJu**
>
> ### **W4: Differentiation from prior in-context segmentation methods.**
>
> We respectfully clarify that iCAS is not merely a technical variation of existing generic segmentation models (SAM, SegGPT, SINE), but a specialized framework designed from the ground up to address the unique constraints and challenges of industrial anomaly detection.
>
> To explicitly demonstrate the fundamental differences in architecture, training paradigms, and core capabilities, we provide a detailed comparison with related foundational models.
>
> | Method | Base Architecture | Model Component | Training Paradigm | Semantic Anomaly Seg. | Interactive Anomaly Seg. | Few-shot Anomaly Det. | Open-set Anomaly Det. |
> | :--- | :--- | :--- | :--- | :--- | :--- | :--- | :--- |
> | SAM | ViT-H | Mask Decoder | SA-1B (11,000K) | ✗ | ✓ | ✗ | ✗ |
> | SegGPT | Painter (ViT-L) | Linear Head | Painter+ADE20K+COCO+VOC+Cityscapes… (162K+273K) | ✓ | ✗ | ✗ | ✗ |
> | HQ-SAM | SAM (ViT-H) | HQ-Output Token and MLPs | SA-1B+HQSeg-44K (11,000K+44K) | ✗ | ✓ | ✗ | ✗ |
> | Matcher | SAM (ViT-H) + DINOv2 (ViT-L) | PLM+RPS+ILM | SA-1B (11,000K) | ✗ | ✗ | ✗ | ✗ |
> | SINE | DINOv2 (ViT-L) | Mask Decoder | ADE20K+COCO+Obj365 (776K) | ✓ | ✓ | ✗ | ✗ |
> | **iCAS (Ours)** | DINOv2 (ViT-L) | Mask Decoder + In-context Queries + GQS Tokens + MFM Module | **General-to-Specific** (861K) | ✓ | ✓ | **✓** | **✓** |
>
> As shown in the Table, prior approaches struggle in standard anomaly detection settings. Whether in Unsupervised Few-shot AD (where the reference is Normal, and the model must find deviations) or Few-shot AD with Anomalies (where the reference contains specific defects), generic models fail because they are designed to learn the semantic alignment without distinguishing between normal and abnormal variations. iCAS is specifically trained to bridge this gap.
> Besides, We propose a completely different set of model components tailored to solve the anomaly segmentation problem. These components collectively form a new reasoning mechanism that enables iCAS to outperform generic baselines significantly.
>
> ### **W5: Comparison with existing V-L anomaly detection models.**
>
> We respectfully clarify the comparison context regarding backbone scale, model capabilities, and functional objectives. There may be some misunderstanding regarding the source of iCAS’s performance gains. The VL baselines (WinCLIP, AnomalyCLIP, InCTRL, etc.) rely on CLIP/OpenCLIP backbones pre-trained on the massive WIT-400M dataset with 400 million image–text pairs. In contrast, iCAS is built on the DINOv2 backbone, pre-trained on a substantially smaller dataset (LVD-142M) and further trained on only 861K task-specific samples. Therefore, the improvements achieved by iCAS reflect higher data efficiency rather than benefits from larger-scale pre-training.
>
> More importantly, the distinction between iCAS and VL-based models lies in functional objectives and industrial applicability, rather than backbone choice alone. VL models are designed to leverage language for zero-shot or few-shot anomaly detection, which constrains their functionality to tasks that can be expressed in text. Even with access to massive pre-training data, they are typically limited to classification or coarse localization, making them less suitable for fine-grained segmentation of novel or undefined industrial defects.
>
> iCAS, on the other hand, is motivated by real-world industrial inspection needs. Its design explicitly considers task complexity and scalability:
>
> **Few-shot Anomaly Detection**: Supports online anomaly detection in new scenarios with only a few samples, enabling rapid adaptation to previously unseen environments.
>
> **Interactive Anomaly Segmentation**: Provides tools for interactive segmentation of real defects, allowing annotators to handle challenging cases where unsupervised or few-shot methods cannot meet industrial quality standards.
>
> **Semantic Anomaly Segmentation & Open-set Anomaly Detection**: Facilitates efficient use of limited anomalous data, allowing iterative enhancement of anomaly detection models and extending capabilities to novel or undefined defects.
>
> By addressing these industrial requirements, iCAS is capable of scaling across diverse tasks and scenarios, providing functionalities that generic VL-based models typically cannot achieve. This task-driven design, combined with a curated General-to-Specific training strategy, ensures practical applicability, pixel-level precision, and adaptability to complex real-world scenarios.

---

> ### Author Response · Authors · 2025-11-26
> **(4/4) Response to Reviewer AgJu**
>
> ### **W6: Novelty of the General-to-Specific pre-training scheme.**
>
> We respectfully point out that there is a fundamental misunderstanding regarding the training paradigms of the mentioned methods. iCAS is the only framework that combines General Pre-training and Anomaly-Specific Pre-training (on source domains) to enable Training-free adaptation on the target domain.
>
> To clarify the distinct mechanisms, we provide a detailed comparison of training strategies and functionalities:
>
> | Method | General Pre-training | Anomaly Pre-training (Source) | Target Training-Free | Few-shot AD | Open-set AD | Interactive Seg. | Semantic Seg. |
> | :--- | :--- | :--- | :--- | :--- | :--- | :--- | :--- |
> | RegAD | ✗ | ✗ | ✗ | ✓ | ✗ | ✗ | ✗ |
> | MetaUAS | ✓ | ✗ | ✓ | ✓ | ✗ | ✗ | ✓ |
> | RealNet | ✗ | ✗ | ✗ | ✗ | ✗ | ✗ | ✗ |
> | **iCAS (Ours)** | ✓ | **✓** | **✓** | **✓** | **✓** | **✓** | **✓** |
>
> **Difference from RegAD (Target Fine-tuning vs. Source Pre-training)**
>
> * RegAD is not a pre-training method in the context of generalist models. It essentially performs fine-tuning (or registration) on the target category using few-shot normal samples. It does not undergo general-purpose visual training.
> * In contrast, iCAS is trained entirely on irrelevant source datasets (General + Anomaly Source) and is applied to the target domain without any parameter updates, achieving true in-context generalization.
>
> **Difference from MetaUAS (Missing the "Specific" Stage)**
> * MetaUAS uses synthetic image pairs for meta-learning, but it does not incorporate an anomaly-specific pre-training stage.
> * Our method, in contrast, first trains on general data to obtain a class-agnostic mask set, and then fine-tunes this mask set using anomaly-specific data to enhance the perception of anomaly boundaries.
>
> **Difference from RealNet (Full-shot AD vs. In-Context AD)**
>
> * RealNet operates on a completely different paradigm aiming for full-shot unsupervised training. It requires the entire set of normal samples from the target domain (e.g., full MVTecAD train set) to synthesize anomalies and train a reconstruction model. It's not aim to the few-shot or online setting.
> * iCAS, conversely, targets few-shot normal and anomalous scenarios where the target dataset is unavailable.
>
>
> Overall, the "General-to-Specific" scheme in iCAS is a unique pipeline. It is a strategic design to bridge the gap between generic segmentation (general) and anomaly discovery (specific) within a unified, training-free inference framework. This specific combination of strategies and supported functionalities is unique and distinct from the baselines mentioned.
>
> ### Summary:
> **We sincerely appreciate Reviewer AgJu's comprehensive suggestions and evaluation. Based on our analysis above, we have enriched the paper in blue font with the following additions:**
>
> 1. **We have incorporated theoretical analysis in Appendix A.6 and A.7.**
> 2. **We have supplemented Appendix A.5 with detailed training, inference, and model comparisons against other in-context segmentation methods.**
> 3. **We have included analysis and discussion of specialized training approaches for anomaly segmentation in Lines 295–303.**

---

### Meta-Review · Area_Chair_E6ba · 2026-01-06

**Summary:**

This paper proposes iCAS, an “in-context” anomaly segmentation framework for industrial inspection built on a mask-classification transformer with two modules—Greedy Query Selection (GQS) and Mask-level Feature Matching (MFM)—plus a General-to-Specific training scheme (general semantic segmentation data followed by anomaly-focused data). The reviewers generally agree that the work is empirically strong: broad experiments across multiple benchmarks, consistent gains over several promptable/in-context baselines, and ablations showing contributions from each component.

However, the meta-level concerns that determine the recommendation are about conceptual contribution, claim accuracy, and methodological positioning. In particular:
- The paper’s central framing of “training-free” / “in-context reasoning” was viewed by one reviewer as overstated given the reliance on substantial supervised pretraining and anomaly-specific training.
- Multiple reviewers raise that the method appears to be an engineering integration of known ideas, with limited conceptual novelty beyond assembling and tuning components.
- There were also concerns about fairness of comparisons (e.g., baselines trained on different data), missing comparisons to strong few-shot anomaly methods, and lack of backbone diversity / sensitivity studies.

The rebuttal is substantive and adds several targeted experiments . While these additions meaningfully strengthen the empirical case and partially address fairness/coverage gaps, they do not fully resolve the core issue that the paper’s main contribution is incremental and that the “in-context / training-free” claims risk being primarily a rebranding of a pretrained promptable segmentation model with anomaly-focused fine-tuning.

Given two below-threshold scores (AgJu: 4; eNU5: 4) and the most positive reviewer still characterizes novelty as “moderate,” I do not think the paper clears the ICLR acceptance bar on conceptual contribution and framing, despite strong experiments.

**Reviewer Concerns:**

**A) Claim accuracy and conceptual framing (“training-free”, “in-context reasoning”, novelty)**
- Addressed
  - The authors clarified that “training-free” refers to no per-target fine-tuning at inference (post pretraining), and argued this aligns with common foundation-model usage.
  - They provided a more explicit differentiation table vs SAM/SegGPT/HQ-SAM/Matcher/SINE and argued iCAS supports more anomaly-specific tasks.
- Outstanding
  - The main criticism remains: iCAS is not training-free in the standard anomaly detection sense, and “in-context reasoning” may be viewed as prompt-conditioned inference of a trained segmentation model rather than a new reasoning mechanism. The rebuttal clarifies terminology but does not eliminate the perception that the paper’s central novelty is mostly integration + training recipe rather than a new modeling principle.

**B) Comparisons and fairness (missing baselines, unequal data/training, VL-AD models)**
- Addressed
  - Added comparisons to recent few-shot anomaly methods DictAS and UniVAD with quantitative results.
  - Responded to fairness concerns vs PerSAM/Matcher by adding medical dataset comparisons where iCAS was not trained on those domains; also provided parameter/data-scale comparisons and argued iCAS is more lightweight.
- Outstanding
  - Reviewer AgJu’s concern that gains may reflect backbone choice and training scale rather than formulation is only partially addressed. The authors argue iCAS is more data-efficient than CLIP-based VL models, but the paper still does not provide a clean apples-to-apples comparison under matched training data and backbone families across model classes.
  - The added DictAS/UniVAD results help, but also reveal metric-dependent behavior, leaving open what the “right” comparison target is for the paper’s multi-task claims.

**C) Technical clarity, internal mechanism explanation, and theoretical grounding of modules**
- Addressed
  - The rebuttal includes a more detailed explanation of the in-context transformer.
  - The authors provide a theoretical motivation for GQS via a k-center formulation with a 2-approx guarantee, and explain why class-conditional or random queries are suboptimal for sparse, unknown anomalies.
  - Provided additional ablations on K (number of queries) and reported stable performance.
- Outstanding
  - While the k-center framing improves interpretability, it still does not clearly establish that GQS/MFM introduce fundamentally new capability beyond more efficient query coverage and mask-level similarity scoring. The modules remain plausible but incremental, and the paper’s narrative may still overstate them as a “new reasoning mechanism.”

**D) Generalization beyond industrial defects and robustness across domains**
- Addressed
  - The authors added experiments on three medical anomaly segmentation datasets, showing sizable gains vs SAM, which strengthens the claim that the approach is not purely industrial-surface-specific.
  - eNU5 also asked for robustness beyond industrial datasets; the medical results partially satisfy this.
- Outstanding
  - The new medical experiments are encouraging, but the evaluation is still narrow in terms of broader natural images / open-world anomalies, and the paper does not yet convincingly separate what generalizes because of (i) DINOv2 features, (ii) the mask transformer training, (iii) anomaly-specific fine-tuning, or (iv) the prompting design.

**E) Efficiency and computational cost**
- Addressed
  - Authors report inference time comparisons and provide an analysis of training iterations and frozen-backbone strategy vs SAM’s full training.
  - Additional cost breakdown and “data efficiency” arguments were added.
- Outstanding
  - Training cost remains non-trivial (multi-stage training, 3.6M iterations), and the cost comparison to other anomaly-specific frameworks (not only SAM family) remains incomplete.

**Reviewer Scores:**

- `AgJu (4, confidence 3): likely 4 → 4 `. The rebuttal directly targets this reviewer’s objections with stronger theoretical framing for GQS/MFM and more explicit differentiation tables. Still, the reviewer’s core stance is that novelty is limited and the “in-context/training-free” framing is overstated. This likely prevents crossing clearly into accept.
- `QgxX (6, confidence 4): likely remains 6.` Their main questions were addressed with new experiments and efficiency tables.
- `eNU5 (4, confidence 3): likely 4 → 6.` The authors addressed missing baselines, backbone ablations, GQS K-sensitivity, and cost analysis. This could lift the score to borderline accept.
- `Tqxe (6, confidence 4): likely remains 6.` Concerns about motivation, clarity, fairness vs PerSAM/Matcher, and missing details were addressed with expanded explanation and additional experiments. No obvious reason for a downgrade. It may stay positive.

The panel likely becomes mixed but closer. Nonetheless, the acceptance decision still hinges on whether the community views the contribution as a sufficiently novel ML contribution rather than a well-engineered training pipeline for anomaly segmentation, and the remaining novelty/claim-framing concerns keep me on the reject side.

---

### Decision · Program_Chairs · 2026-01-26

Reject